

Atmospheric
Chemistry
and Physics

# The impact of organic pollutants from Indonesian peatland fires on the tropospheric and lower stratospheric composition

**Simon Rosanka**[1], **Bruno Franco**[2], **Lieven Clarisse**[2], **Pierre-François Coheur**[2], **Andrea Pozzer**[3], **Andreas Wahner**[1], and **Domenico Taraborrelli**[1]

[1]TS1 Institute of Energy and Climate Research: Troposphere (IEK-8),
Forschungszentrum Jülich GmbH, 52425 Jülich, Germany
[2]Spectroscopy, Quantum Chemistry and Atmospheric Remote Sensing (SQUARES),
Université libre de Bruxelles (ULB), Brussels 1050, Belgium
[3]Atmospheric Chemistry Department, Max-Planck-Institute for Chemistry, 55128 Mainz, Germany

**Correspondence:** Simon Rosanka (s.rosanka@fz-juelich.de)

**Abstract.** TS2 The particularly strong dry season in Indonesia in 2015, caused by an exceptionally strong El Niño, led to severe peatland fires resulting in high volatile organic compound (VOC) biomass burning emissions. At the same time, the developing Asian monsoon anticyclone (ASMA) and the general upward transport in the Intertropical Convergence Zone (ITCZ) efficiently transported the resulting primary and secondary pollutants to the upper troposphere and lower stratosphere (UTLS). In this study, we assess the importance of these VOC emissions for the composition of the lower troposphere and the UTLS and investigate the effect of in-cloud oxygenated VOC (OVOC) oxidation during such a strong pollution event. This is achieved by performing multiple chemistry simulations using the global atmospheric model ECHAM/MESSy (EMAC). By comparing modelled columns of the biomass burning marker hydrogen cyanide (HCN) and carbon monoxide (CO) to spaceborne measurements from the Infrared Atmospheric Sounding Interferometer (IASI), we find that EMAC properly captures the exceptional strength of the Indonesian fires.

In the lower troposphere, the increase in VOC levels is higher in Indonesia compared to other biomass burning regions. This has a direct impact on the oxidation capacity, resulting in the largest regional reduction in the hydroxyl radical (OH) and nitrogen oxides ($NO_x$). While an increase in ozone ($O_3$) is predicted close to the peatland fires, simulated $O_3$ decreases in eastern Indonesia due to particularly high phenol concentrations. In the ASMA and the ITCZ, the up-ward transport leads to elevated VOC concentrations in the lower stratosphere, which results in the reduction of OH and $NO_x$ and the increase in hydroperoxyl radical ($HO_2$). In addition, the degradation of VOC emissions from the Indonesian fires becomes a major source of lower stratospheric nitrate radicals ($NO_3$), which increase by up to 20 %. Enhanced phenol levels in the upper troposphere result in a 20 % increase in the contribution of phenoxy radicals to the chemical destruction of $O_3$, which is predicted to be as large as 40 % of the total chemical $O_3$ loss in the UTLS. In the months following the fires, this loss propagates into the lower stratosphere and potentially contributes to the variability of lower stratospheric $O_3$ observed by satellite retrievals. The Indonesian peatland fires regularly occur during El Niño years, and the largest perturbations of radicals concentrations in the lower stratosphere are predicted for particularly strong El Niño years. By activating the detailed in-cloud OVOC oxidation scheme Jülich Aqueous-phase Mechanism of Organic Chemistry (JAMOC), we find that the predicted changes are dampened. Global models that neglect in-cloud OVOC oxidation tend to overestimate the impact of such extreme pollution events on the atmospheric composition.

# 1  Introduction

Particularly strong Indonesian wildfires during the El Niño in 2015 led to severe air pollution and reduced visibility (Kim et al., 2015; Lee et al., 2017), resulting in increased morbidity and mortality (Marlier et al., 2013; Reddington et al., 2014; Crippa et al., 2016) in South East Asia (SEA). In general, El Niño is a large-scale climate anomaly, which is characterised by significantly warmer eastern equatorial Pacific Ocean sea surface temperatures (Trenberth, 1997), resulting in a dry season in SEA (Weng et al., 2007). The very strong El Niño phase in 2015–2016, which is the third strongest on record (after 1997–1998 and 1982–1983, NOAA, 2020), led to a particularly strong dry season in Indonesia (Jiménez-Muñoz et al., 2016), which started in mid-July and lasted until November (Field et al., 2016). In the past, much of the originally forested and moist peatland in Kalimantan and Sumatra has been drained and cleared during agricultural land management. In order to clear these forests, landscape fires are commonly used. Even small local fires in these regions during non-El Niño years may induce particularly strong biomass burning emissions. Gaveau et al. (2014) estimated that a local 1-week Indonesian biomass burning event in 2013 contributed to about 5 %–10 % of Indonesian's total greenhouse gas emissions in that year. The additional drying during El Niño years favours fires that burn deep down into the peat and can last for multiple weeks. Due to their long lifetimes, these fires spread and ignite new areas, which are not necessarily prone to biomass burning. Compared to non-El Niño years, this results in strong biomass burning emissions from Indonesia (van der Werf et al., 2017). The underground conditions inherently determine smouldering fires, which are characterised by low combustion temperatures. In combination with the high carbon content of peat, smouldering fires emit much larger amounts of non-$CO_2$ emissions from peatlands than from other fuels (Christian et al., 2003; Rein et al., 2009; Yu et al., 2010). A major fraction of these non-$CO_2$ emissions is volatile organic compounds (VOCs), which comprise a large variety of species and can influence atmospheric chemistry on a regional and global scale. In the atmosphere, VOCs mainly react with the hydroxyl radical (OH), ozone ($O_3$), and the nitrate radical ($NO_3$) or photodissociate. Their atmospheric lifetimes range from minutes to years. Figure 1 shows the dry matter burned (DMB) during the 2015 Indonesian fires along the distribution of the peatlands (indicated in blue). It becomes evident that most of the areas influenced by biomass burning (e.g. Sumatra, Kalimantan) are covered with peatland, indicating that the 2015 Indonesian fires are characterised by high VOC emissions.

During the Indonesian biomass burning season, usually the Asian monsoon is ongoing such that a large anticyclone spanning from tropical to temperate regions (from about 10 to 40° N) evolves. This semi-stationary large-scale meteorological pattern typically extends from the Middle East to Asia in the upper troposphere and lower stratosphere (UTLS) (Basha

et al., 2020). As a convective system, the Asian monsoon anticyclone (ASMA) acts as a pollution pump facilitating a fast transport of surface emissions to the UTLS (Park et al., 2008; Randel et al., 2010; Lelieveld et al., 2018). Vogel et al. (2015) analysed the impact of different regions in Asia on the chemical composition of the 2012 ASMA by using a chemical Lagrangian model. They found that air masses from SEA contribute significantly to the composition of the anticyclone in the UTLS. In addition, the vertically convective transport in the Intertropical Convergence Zone (ITCZ) and in the southeastern flank of the anticyclone carries air masses from SEA into the UTLS. Thus, even short-lived VOCs from Indonesian fires are transported into the UTLS and potentially affect the lower stratospheric composition.

The Asian monsoon is characterised by the frequent occurrence of clouds and precipitation, and it has been demonstrated that the ASMA has a higher water vapour content than other meteorological systems (Fu et al., 2006). At the same time, the Madden–Julian Oscillation (MJO) leads to enhanced water vapour concentrations and precipitation over the Indian Ocean and Indonesia (Zhang, 2013). Many oxygenated VOCs (OVOCs) have a high solubility and quickly partition and react in cloud droplets influencing radical concentrations and the atmospheric composition in general (Herrmann et al., 2015). Rosanka et al. (2020b) showed that the in-cloud OVOC oxidation has a significant impact on the predicted concentrations of VOCs, key oxidants, and $O_3$. In the past, global atmospheric chemistry models were not capable of representing this process explicitly or in its full complexity (Ervens, 2015). However, the recently developed Jülich Aqueous-phase Mechanism of Organic Chemistry (JAMOC, Rosanka et al., 2020c, b) comprises an advanced in-cloud OVOC oxidation scheme suitable to be used in the ECHAM/MESSy Atmospheric Chemistry (EMAC, Jöckel et al., 2010) model. This allows us to assess the importance of this in-cloud oxidation process during the VOC-dominated Indonesian peatland fires.

In this study, we therefore investigate the importance of biomass burning VOC emissions from the strong 2015 Indonesian peatland fires on (1) the lower tropospheric composition, (2) the UTLS, and (3) the importance of in-cloud OVOC oxidation in such an extreme pollution event. This is addressed by performing multiple global chemistry simulations using the ECHAM/MESSy Atmospheric Chemistry (EMAC, Sect. 2) model. In addition to the 2015 fires, strong peatland fires frequently occur in Indonesia. Especially during El Niño years (in 2002–2003, 2004–2005, 2006–2007, 2009–2010, and 2014–2016), high emissions have been observed (van der Werf et al., 2017). Therefore, the long-term impact of these periodically occurring events is additionally addressed. Globally, biomass burning is not limited to Indonesia, and many regions are frequently affected. In each region, biomass burning varies in strength, frequency, the characteristics of the biomass burned, and the chemical background conditions. In a first step, we therefore compare the

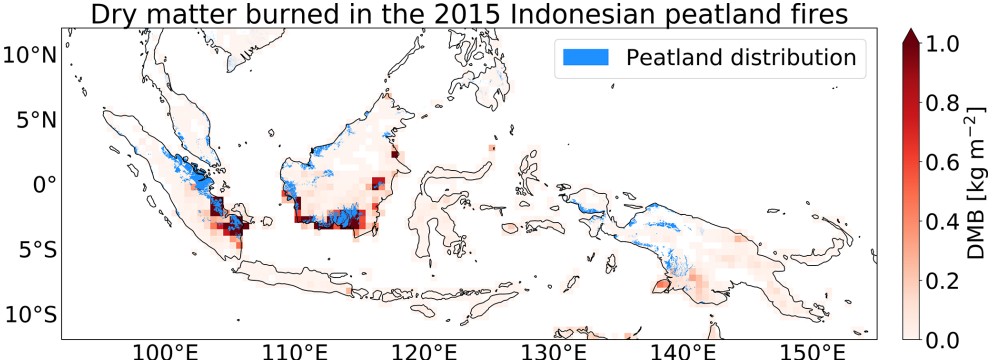

**Figure 1.** Accumulated dry matter burned (DMB) during the Indonesian peatland fires of 2015. The distribution of Indonesian peatland is indicated in blue. The data for the peatland distribution are obtained from Xu et al. (2017, 2018).

Indonesian peatland fires to other biomass burning regions, focusing on their specific emission footprint (Sect. 3). The ability of EMAC to represent biomass burning events is evaluated using hydrogen cyanide (HCN) and carbon monoxide (CO) satellite retrievals (Sect. 4). Afterwards the impact of the 2015 Indonesian peatland fires on the troposphere and the UTLS is analysed, focusing on hydrocarbons, oxygenated organics, nitrogen-containing compounds, key radicals, and $O_3$ in Sects. 5 and 6, respectively. In Sect. 7, the importance of in-cloud OVOC oxidation during this pollution event is addressed. Modelling uncertainties related to this study are discussed in Sect. 8 before drawing final conclusions (Sect. 9).

## 2 Modelling approach

This section provides an overview on the global model used in this study. The main focus is placed on the representation of atmospheric gas- and aqueous-phase chemistry, biogenic and biomass burning emissions, and the strategy to compare EMAC's simulated results with satellite retrievals (Sect. 2.1). Section 2.2 provides an overview of each simulation performed in this study.

### 2.1 EMAC

The ECHAM/MESSy Atmospheric Chemistry (EMAC) model is a numerical chemistry and climate simulation system that includes submodels describing tropospheric and middle-atmosphere processes and their interaction with oceans, land, and human influences (Jöckel et al., 2010). It uses the second version of the Modular Earth Submodel System (MESSy2) to link multi-institutional computer codes. The core atmospheric model is the fifth-generation European Centre Hamburg general circulation model (ECHAM5, Roeckner et al., 2006). Jöckel et al. (2010) provided an update on all modelling components used. For the present study, we applied EMAC (ECHAM5 version 5.3.02, MESSy version 2.54.0) in the T106L90MA and T42L90MA resolution, i.e. with a spherical truncation of T106 and T42 (cor-

responding to a quadratic Gaussian grid of approximately 1.1° by 1.1° and 2.8° by 2.8°, respectively). By using this horizontal resolution, addressing the short-term implications for 2015–2016 as well as the long-term impact (2001–2016) on a global scale is still feasible while at the same time the computational costs are affordable. For both resolutions, 90 (L90) vertical hybrid pressure levels up to 0.01 hPa are used focusing on the lower and middle atmosphere (MA), representing tropospheric and stratospheric transport processes reasonably well (Jöckel et al., 2010). Thus, the impact on the troposphere and the UTLS can be addressed. A detailed discussion on the comparability of both resolutions is performed in Sect. 8.

### 2.1.1 Atmospheric chemistry

For this study, the gas- and aqueous-phase chemical kinetics is integrated by two separate submodels. For the atmospheric gas-phase chemistry, the applied model setup comprised the submodel Module Efficiently Calculating the Chemistry of the Atmosphere (MECCA, Sander et al., 2019) using the gas-phase Mainz Organic Mechanism (MOM). MOM contains an extensive oxidation scheme for isoprene (Taraborrelli et al., 2009, 2012; Nölscher et al., 2014), monoterpenes (Hens et al., 2014), and aromatics (Cabrera-Perez et al., 2016) and is therefore capable of representing all the biomass burning VOCs considered in EMAC. In addition, comprehensive reactions schemes are considered for the modelling of the chemistry of $NO_x$ ($NO + NO_2$), $HO_x$ ($OH + HO_2$), $CH_4$, and anthropogenic aliphatic and aromatic hydrocarbons. VOCs are oxidised by OH, $O_3$, and $NO_3$, whereas peroxy radicals ($RO_2$) react with $HO_2$, $NO_x$, and $NO_3$ and undergo self- and cross-reactions (Sander et al., 2019). Isocyanic acid (HNCO) is a chemical constituent that is primarily emitted by biomass burning and potentially harmful to humans (Wang et al., 2007; Roberts et al., 2011; Leslie et al., 2019). In order to properly represent this toxic constituent within EMAC, MOM has been extended to represent the atmospheric chemistry of HNCO. For this, the mecha-

nism proposed by Rosanka et al. (2020d) is implemented into MOM. Their mechanism includes formamide as an additional chemical source of HNCO and chemical mechanisms for nitromethane, methylamine, dimethylamine, and trimethylamine.

The atmospheric aqueous-phase chemistry is modelled using the SCAVenging submodel (SCAV, Tost et al., 2006). It simulates the removal of trace gases and aerosol particles by clouds and precipitation. SCAV calculates the transfer of species into and out of rain and cloud droplets using the Henry's law equilibrium, acid dissociation equilibria, oxidation–reduction reactions, heterogeneous reactions on droplet surfaces, and aqueous-phase photolysis reactions (Tost et al., 2006). As mentioned earlier and as demonstrated by Rosanka et al. (2020b), in-cloud OVOC oxidation significantly influences the atmospheric composition. However, the ordinary differential equations (ODE) systems resulting from the combination of gas-phase and in-cloud aqueous-phase suffer from (1) a higher stiffness due to fast acid–base equilibria and phase-transfer reactions and (2) load imbalance on high-performance computing (HPC) systems due to the sparsity of clouds. This leads to a significant increase in computational costs when using larger chemical mechanisms like the Jülich Aqueous-phase Mechanism of Organic Chemistry (JAMOC), i.e. larger ODE systems (Rosanka et al., 2020c). Using JAMOC in each simulation performed in this study is thus not feasible. As a trade-off, JAMOC is used in a simulation subset in order to address and estimate its implications on the other simulations. Thus, two different aqueous-phase mechanisms are used within this study: (1) the standard aqueous-phase mechanism of EMAC (in the following called ScSta), which includes a detailed oxidation scheme and represents more than 150 reactions (Jöckel et al., 2016), and (2) JAMOC (Rosanka et al., 2020c), which includes a complex in-cloud OVOC oxidation scheme. In JAMOC, the phase transfer of species containing up to 10 carbon atoms and the oxidation of species containing up to 4 carbon atoms are represented. Similar to MOM, both aqueous-phase mechanisms are modified to include the changes proposed by Rosanka et al. (2020d) to properly represent HNCO.

### 2.1.2   Biogenic and biomass burning VOC emissions

In the atmosphere, biogenic and biomass burning emissions are two major sources of VOCs. The largest biogenic emissions take place in the equatorial region (e.g. Amazon basin, central Africa) with additional emissions in the Northern Hemisphere (NH) and Southern Hemisphere (SH) extratropics. The MESSy submodel uses the Model of Emissions of Gases and Aerosols from Nature (MEGAN, Guenther et al., 2006) to calculate biogenic VOC emissions. The global emissions of isoprene, the most abundant biogenic VOC, are scaled to $595\,\mathrm{Tg\,a^{-1}}$, the best estimate of Sindelarova et al. (2014).

Biomass burning emission fluxes are calculated using the MESSy submodel BIOBURN, which determines these fluxes based on biomass burning emission factors and dry matter combustion rates. For the latter, data from the Global Fire Assimilation System (GFAS) that are based on satellite observations of the fire radiative power obtained from the Moderate Resolution Imaging Spectroradiometer (MODIS) satellite instruments are used (Kaiser et al., 2012). In BIOBURN, the emission strength depends on the dominant fuel type in the respective area. From the GFAS dataset used in EMAC, in 2015, the dominant fuel type over Indonesia is tropical forest fire. However, as discussed earlier, peatland fires contribute substantially to the Indonesian fires. The GFAS dataset of EMAC is changed such that the dominant fuel type over Indonesia is a combination of peat and tropical forest fires with equal contributions (following van der Werf et al., 2017). In general, biomass burning emission factors for VOCs are based on Akagi et al. (2011). Biomass burning emissions for HNCO, formamide, nitromethane, methylamine, dimethylamine, and trimethylamine are implemented following Rosanka et al. (2020d) using emission factors from Koss et al. (2018) for HNCO and formamide.

### 2.1.3   Observational comparison

The evaluation of model simulation results against global observational datasets of VOC abundance can be performed for only a few species, mainly because of the limited availability in spaceborne measurements of such compounds. Among them, several VOCs are retrieved globally from the observations made by the nadir-viewing hyperspectral Infrared Atmospheric Sounding Interferometer (IASI, Clerbaux et al., 2009). Embarked on the Metop platforms on sun-synchronous polar orbits, IASI crosses the Equator at 09:30 and 21:30 local solar time and achieves a global coverage twice daily with a fairly dense spatial sampling. Here, we make use of the HCN abundance retrieved from the IASI/Metop-A and B observations to assess the ability of EMAC to represent such an important biomass burning event. In addition, IASI methanol ($CH_3OH$) data are used to assess the impact of in-cloud OVOC oxidation in the model simulations (Sect. 7).

The retrieval method used to obtain the HCN measurements from the IASI observations follows closely the version 3 of the Artificial Neural Network for IASI (ANNI), which already allowed the retrieval of a suite of VOCs, including $CH_3OH$ (Franco et al., 2018). ANNI is a general retrieval framework that consists in quantifying, for each IASI observation, the spectral signature of the target gas with a sensitive hyperspectral metric and in converting this metric into gas total column via an artificial feedforward neural network (NN). Details on the ANNI retrieval approach, the HCN retrieval specificities, and the HCN product itself are provided in Appendix A. We refer to Franco et al. (2018) for a description of the IASI methanol retrievals. The satel-

lite datasets exploited in this study consist of daily global distributions of HCN and $CH_3OH$ total columns derived from the daytime observations (approximately 09:30 local time) of the IASI/Metop-A and B overpasses. These offer a better measurement sensitivity than the evening overpasses (Franco et al., 2018). Scenes affected by clouds or poor retrieval performance are removed from the final dataset by specific filters. Examples of daily regional distributions of HCN columns in the 2015 Indonesian fires as well as the seasonal global distributions of HCN as retrieved from IASI are presented in Appendix A. Those highlight the ability of IASI to capture the enhancements of HCN during biomass burning events as well as its downwind transport over long distances.

Significant enhancements of carbon monoxide (CO) have already been captured by IASI in the 2015 Indonesian fires (e.g. Whitburn et al., 2016b; Nechita-Banda et al., 2018). Therefore, we also evaluate the ability of EMAC to reproduce the CO columns observed from space during this event. The vertical profile and column abundance of CO are obtained in near real time from the IASI/Metop-A and B spectra with the Fast Optimal Retrievals on Layers for IASI (FORLI) algorithm (Hurtmans et al., 2012). Several quality flags ensure that IASI observations affected by clouds, unstable retrieval, and measurement sensitivity that is too weak are excluded from the final CO dataset. The FORLI algorithm, characterisation of the retrieved CO product, and validation against independent measurements are reported in Hurtmans et al. (2012) and George et al. (2015). Following the formalism of Rodgers (2000), the IASI averaging kernels are applied to the CO model profiles to account for the inhomogeneous vertical sensitivity of the IASI measurements and to compute modelled CO columns as would be seen by the satellite instrument (see e.g. Sect. 5.1 in Schultz et al., 2018).

## 2.2 Simulations performed

Within this study, seven simulations are performed, which can be summarised in three simulation sets. Each simulation differs either in the biomass burning emissions, the aqueous-phase mechanism used, or the modelled time period. Table 1 provides an overview of all simulations and their characteristics. For each simulation set, in one simulation all VOC emissions from biomass burning are switched off (named REF and $REF_{LONG}$). A second simulation includes biomass burning VOC emissions as described in Sect. 2.1.2 (named FIR and $FIR_{LONG}$). Performing high-resolution simulations with the highest complexity in the chemical mechanisms in EMAC comes with high computational costs. The strong Indonesian peatland fires of 2015 and the following year are selected as a specific case study (named REF and FIR). For both simulations, the year 2014 is simulated as spin-up, which is not considered for the analysis. For this case study, high-resolution simulations are performed at T106L90MA. In order to isolate the impact of the In-

donesian peatland fires in 2015, an additional simulation (named $FIR_{NOINDO}$) is performed, for which all biomass burning VOC emissions from Indonesia are switched off. In order to address the impact of in-cloud OVOC oxidation on such a VOC-dominated pollution event, two simulations including JAMOC are performed (named $REF_{JAMOC}$ and $FIR_{JAMOC}$). However, to reduce the computational demand (see Sect. 2.1.1), these simulations focus only on the second half of 2015 at T106L90MA resolution. The long-term effect of reoccurring Indonesian peatland fires are addressed by performing two long simulations for the time period of 2001–2016 (named $REF_{LONG}$ and $FIR_{LONG}$). Here, the year 2000 is simulated for spin-up, which is not used for the analysis. Performing these simulations at T106L90MA and using JAMOC is computationally not feasible. Therefore, the EMAC's standard aqueous-phase mechanism is used and the resolution is reduced to T42L90MA. All simulations are performed using the chemistry–transport model mode (QCTM mode, Deckert et al., 2011), meaning that chemistry and dynamics are decoupled; e.g. fixed tracer mixing rations are used as input for the radiation scheme instead of the prognostic chemical tracers. In this way, the meteorology is the same for all simulations, and all changes in the atmospheric chemical composition predicted by EMAC are due to either the additional VOC emissions from biomass burning (when comparing REF with FIR or $REF_{LONG}$ with $FIR_{LONG}$) or the in-cloud OVOC oxidation (when comparing REF, FIR, $REF_{JAMOC}$, and $FIR_{JAMOC}$).

## 3 Peatland fires in Indonesia compared to biomass burning in other regions

Globally, biomass burning frequently occurs in seven regions for which Fig. 2 and Table 2 provide an overview. In each region, biomass burning varies in strength, frequency, the characteristics of the biomass burned, and the chemical background conditions. Only about 2.84 % of the Earth's land mass is covered by peatland (Xu et al., 2018), making equatorial Asia the region where the most of peatland is burned. Since non-peatland biomass burning fuels have lower VOC (and higher $NO_x$) emission factors (Akagi et al., 2011), Indonesia is characterised by a unique emission footprint. Figure 3 shows the total trace gas (VOC and non-VOC), the VOC, and the aromatic biomass burning emissions for each region in non-El Niño years, El Niño years, and in 2015 predicted by EMAC (based on $FIR_{LONG}$). In non-El Niño years, the highest total biomass burning emissions of about 1.38 and 1.76 $Pg\,a^{-1}$ originate from central Africa (CAF) and southern Africa (SAF), respectively, whereas the lowest biomass burning emissions of about 0.13 $Pg\,a^{-1}$ occur in Alaska (ALA). SEA contributes only 0.55 $Pg\,a^{-1}$ to the total biomass burning emissions, which is about one-third of the SAF biomass burning emissions. However, in El Niño years this almost doubles (1.05 $Pg\,a^{-1}$), and in the exception-

**Table 1.** List of EMAC simulations performed in this study. Here, ScSta indicates EMAC's standard aqueous-phase mechanism (Jöckel et al., 2016) and JAMOC indicates the complex in-cloud OVOC oxidation scheme by Rosanka et al. (2020c, b) (for further details see Sect. 2.1.1).

| Name | Analysed period | VOC BIOBURN emissions | Aqueous-phase mechanism | Resolution |
|------|-----------------|------------------------|--------------------------|------------|
| REF | 2015–2016 | no | ScSta | T106L90MA |
| FIR | 2015–2016 | yes | ScSta | T106L90MA |
| FIR$_{NOINDO}$ | SOND[a] in 2015 | yes[b] | ScSta | T106L90MA |
| REF$_{LONG}$ | 2001–2016 | no | ScSta | T42L90MA |
| FIR$_{LONG}$ | 2001–2016 | yes | ScSta | T42L90MA |
| REF$_{JAMOC}$ | SOND[a] in 2015 | no | JAMOC | T106L90MA |
| FIR$_{JAMOC}$ | SOND[a] in 2015 | yes | JAMOC | T106L90MA |

[a] Focus on Indonesia in September, October, November, and December. [b] No VOC biomass burning emissions from Indonesian peatland fires.

ally strong year 2015 the biomass burning emissions from SEA of about $1.62\,\mathrm{Pg\,a^{-1}}$ are almost the same as the total biomass burning emissions from SAF. In CAF and SAF, mainly tropical forest and savanna are burned, resulting in low VOC emissions. In 2015, the VOC and aromatic emissions of both regions ranged between 11.32 to $14.34\,\mathrm{Tg\,a^{-1}}$ and 0.89 to $1.20\,\mathrm{Tg\,a^{-1}}$, respectively, which compared to SEA is significantly lower (VOCs: $23.61\,\mathrm{Tg\,a^{-1}}$; aromatics: $2.52\,\mathrm{Tg\,a^{-1}}$). The two northern regions ALA and northern Asia (NAS), which are characterised by extratropical forest with organic soil, add significantly to the global VOC emissions from biomass burning, even though their contribution to the total biomass burning emissions is low. The contribution of NAS to the total biomass burning is less than half of the contribution by SAF ($0.69\,\mathrm{Pg\,a^{-1}}$ compared to $1.76\,\mathrm{Pg\,a^{-1}}$), but its contribution to the aromatic biomass burning emissions is almost the same ($1.11\,\mathrm{Tg\,a^{-1}}$ compared to $1.20\,\mathrm{Tg\,a^{-1}}$). The contribution of ALA strongly depends on the El Niño. In non-El Niño years, the total biomass burning emissions are very low ($0.13\,\mathrm{Pg\,a^{-1}}$) but increase in El Niño years ($0.29\,\mathrm{Pg\,a^{-1}}$). In the exceptionally strong year 2015, the contribution of ALA to the aromatic biomass burning emissions is $0.82\,\mathrm{Tg\,a^{-1}}$, which is of similar strength as from CAF ($0.89\,\mathrm{Tg\,a^{-1}}$) even though its total contribution is only one-third when compared to CAF. The two regions dominated by savanna, central South America (CSA) and northern Australia (NAU) emit 7.72 and $3.30\,\mathrm{Tg\,a^{-1}}$ of VOCs from biomass burning, respectively.

## 4 The representation of the Indonesian peatland fires in EMAC

In order to analyse the ability of EMAC to represent the Indonesian peatland fires, we compare predicted EMAC total columns of HCN and CO to observations obtained from IASI retrievals.

### 4.1 Comparison to IASI HCN retrievals

HCN mainly originates from combustion processes and is therefore largely emitted by biomass burning (Shim et al., 2007). Other emission sources including industrial activities, automobile exhaust, and domestic biofuel are assumed to be very weak (Lobert et al., 1990; Li et al., 2009). Reactions involving acetonitrile ($CH_3CN$) are the only gas-phase source of HCN, but those are estimated to be a minor contribution to the atmospheric HCN burden (Li et al., 2009). The slow oxidation of HCN by OH and $O(^1D)$ is considered to be the most important atmospheric gas-phase sink, leading to long chemical lifetimes (Cicerone and Zellner, 1983). However, due to a strong ocean uptake, the overall atmospheric lifetime is reduced to a few months (Li et al., 2000, 2009). The almost exclusive biomass burning source, combined with a long atmospheric residence time that allows for long-range transport, makes HCN a widely used primary tracer of biomass burning emissions and fire plumes (Li et al., 2009). Moreover, substantial emissions of HCN are expected from strong peatland fires (e.g. Akagi et al., 2011; Andreae, 2019). Therefore, HCN satellite data from IASI are used here to evaluate the performance of EMAC in representing the 2015 Indonesian peatland fires.

At the beginning of the Indonesian fires, the emitted HCN is transported westward, leading to high HCN column values over the Indian Ocean (see Fig. A3). While the fires are ongoing throughout October, the strong westward transport of HCN results in the complete covering of the Indian Ocean. Some HCN is also transported eastward over Australia and the Pacific Ocean. In November, the air masses from Indonesia mix with emissions from Africa and the eastward-transported air masses reach South America. Figure 4 shows the comparison of modelled HCN total columns to IASI satellite retrievals for the 3-month mean with strong peatland emissions in Indonesia. In general, EMAC strongly underestimates HCN when its main source from biomass burning is not taken into account (simulation REF). Once the HCN biomass burning emissions are taken into account,

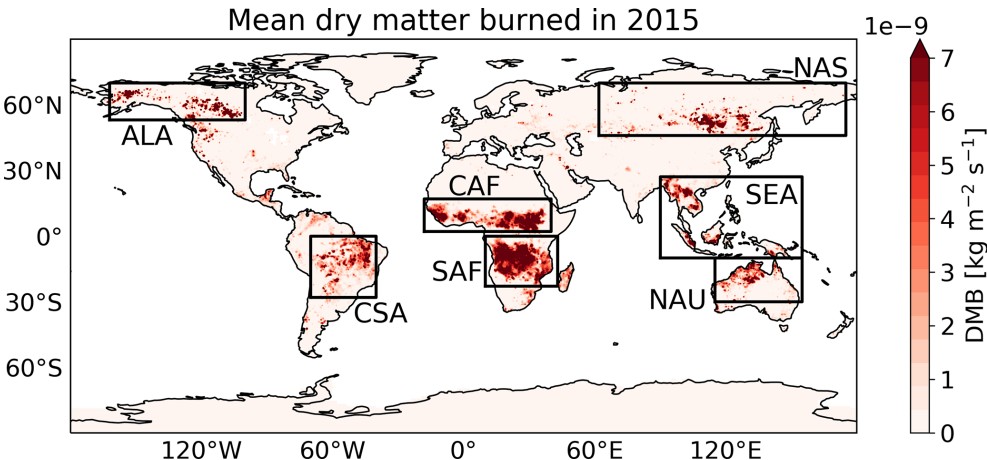

**Figure 2.** Mean dry matter burned (DMB) in 2015. The naming of each region is as follows: Alaska (ALA), central Africa (CAF), central South America (CSA), northern Asia (NAS), northern Australia (NAU), southern Africa (SAF), and South East Asia (SEA). Further details about each region are presented in Table 2.

**Table 2.** Characteristics of the different biomass burning regions focusing on the dominant fuel type, the main biomass burning season, and the dry matter burned (DMB). The global DMB by GFAS (Kaiser et al., 2012) for the year 2015 is $4985\,\mathrm{Tg\,a^{-1}}$. The naming of each region is as follows: Alaska (ALA), central Africa (CAF), central South America (CSA), northern Asia (NAS), northern Australia (NAU), southern Africa (SAF), and South East Asia (SEA). Each region is graphically illustrated in Fig. 2.

| Region | Dominant fuel type | Main biomass burning season | 2015 DMB [$\mathrm{Tg\,a^{-1}}$] |
|--------|--------------------|-----------------------------|-----------------------------------|
| ALA | Extratropical forest with organic soil | JJA | 295 |
| CAF | Tropical forest and savanna | DJF | 778 |
| CSA | Savanna | SON | 439 |
| NAS | Extratropical forest with organic soil | MMA and JJA | 363 |
| NAU | Savanna | SON | 260 |
| SAF | Tropical forest and savanna | JJA | 1036 |
| SEA | Tropical forest[a] | SON | 1237[b] |

[a] In this study a combination of tropical forest (50 %) and peatland (50 %) is assumed in Indonesia (Sect. 2.1.2).
[b] Of which $949\,\mathrm{Tg\,a^{-1}}$ is from Indonesian peatland fires.

the overall underprediction in EMAC is mostly resolved, but EMAC partially overpredicts HCN in SEA. Figure 5 gives the frequency of the global HCN EMAC total column bias in relation to the IASI retrievals during the Indonesian peatland fires, once including biomass burning emissions in the simulations and once not. This comparison clearly shows that HCN is strongly underestimated when its main source is not represented in EMAC. With HCN from biomass burning, the mean column bias reduces from $-5.32 \times 10^{15}$ to $-1.06 \times 10^{15}$ molecules $\mathrm{cm^{-2}}$, and its variance reduces from $1.75 \times 10^{31}$ to $2.57 \times 10^{30}$ molecules$^2$ $\mathrm{cm^{-4}}$, significantly improving the representation of HCN in EMAC.

In general, EMAC's representation of HCN is associated with some uncertainties. Another important source of HCN is terrestrial vegetation, which may contribute to atmospheric concentrations by up to 18 % (Shim et al., 2007). In EMAC, the submodule MEGAN calculates that biogenic emissions contribute about 15 % to the total HCN emissions. Consider-

ing the particularly high atmospheric concentrations in 2015, this slightly lower contribution suggests that this source strength may be well represented in EMAC. Overall, it is expected that the HCN atmospheric lifetime is realistically modelled, since globally HCN columns are well reproduced. Moreover, the ocean uptake accounts for $1.2\,\mathrm{Tg(N)\,a^{-1}}$, which is well in the range of 1.1 to $2.6\,\mathrm{Tg(N)\,a^{-1}}$ proposed by Li et al. (2000) and very close to the Singh et al. (2003) estimate of $1.0\,\mathrm{Tg(N)\,a^{-1}}$. The representation of biomass burning within EMAC depends on satellite observations (Sect. 2.1.2), which retrieve the fire radiative power and are thus sensitive to clouds. This introduces some uncertainties in regions that are characterised by the frequent occurrence of clouds, like equatorial Asia. Focusing on Indonesia, Liu et al. (2020) compared five different global fire inventories and found that GFAS, the inventory used in this study, represents the strength of these fires best. Still, GFAS tends to slightly underestimate the strength, when

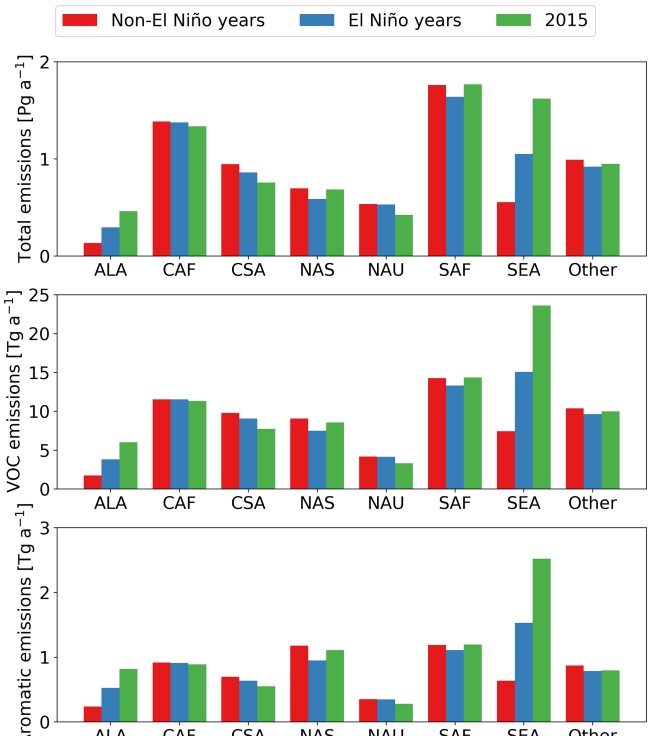

**Figure 3.** The total trace gas (VOC and non-VOC), the VOC, and the aromatic biomass emissions for each region in non-El Niño years, El Niño years, and in 2015 predicted by EMAC (based on REF$_{LONG}$ and FIR$_{LONG}$). Further details about each region are presented in Table 2 and Fig. 2.

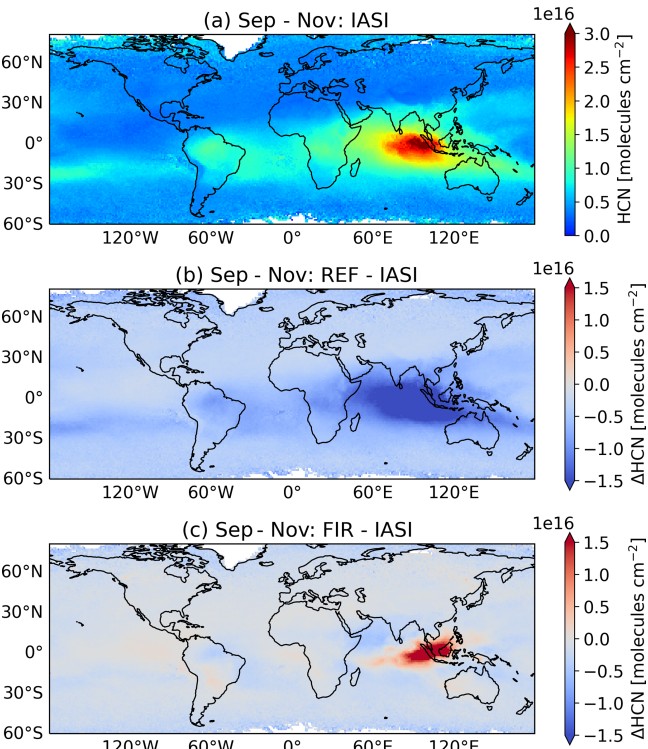

**Figure 4.** HCN comparison between IASI, REF, and FIR. **(a)** Mean global observed IASI HCN columns for September to November. **(b)** Mean global HCN column comparison between REF and IASI for September to November. **(c)** Mean global HCN column comparison between FIR and IASI for September to November.

compared to regional observations in Singapore, Malaysia, and Indonesia. This suggests that the magnitude of the Indonesian fires is well represented in EMAC. However, from the literature a high uncertainty in the emission factors for HCN is reported. Here, we use the emission factors optimised for atmospheric models by Akagi et al. (2011), which suggest $5.0\,\mathrm{g\,kg^{-1}}$ for HCN from peatland fires. From recent field measurements in Indonesia and Malaysia, Stockwell et al. (2016) and Smith et al. (2018) report values ranging from 0.34 to $8.21\,\mathrm{g\,kg^{-1}}$, whereas lab measurements for Indonesian peatland by Stockwell et al. (2015) suggest values between 3.30 and $3.83\,\mathrm{g\,kg^{-1}}$. Overall, this results in a mean emission factor of $4.40\,\mathrm{g\,kg^{-1}}$ across all studies (Andreae, 2019), suggesting that the HCN emission factor used in EMAC is slightly too high, influencing EMAC's overprediction of HCN columns. Lastly, EMAC's overprediction west of Indonesia suggests that some of the overprediction is caused by the deviation of horizontal transport (further discussed in Sect. 8).

## 4.2 Comparison to IASI CO retrievals

At the surface, CO is primarily emitted by natural and anthropogenic combustion processes like biomass and fossil fuel burning and to a lesser extent by biogenic and oceanic sources. The degradation of methane ($CH_4$) and non-methane hydrocarbons (NMHC) in the atmosphere accounts for almost half of the global CO sources (Zheng et al., 2019). In the atmosphere, CO mainly reacts with OH, and the EMAC estimates by Lelieveld et al. (2016) and more recently by Rosanka et al. (2020b) show that CO largely determines the atmospheric oxidation capacity. To a lesser extent, CO is deposited (Stein et al., 2014).

Figure 6a shows the total CO columns observed by IASI for the 3-month mean with strong peatland emissions in Indonesia. Similar to HCN, high CO columns up to about $6.0 \times 10^{18}$ molecules $\mathrm{cm^{-2}}$ are observed over Indonesia. Additionally, high CO columns are also observed in Africa and South America. Compared to HCN, CO is characterised by a shorter lifetime. Therefore, less CO from Indonesia is transported towards Africa at the end of the peatland fire period. Figure 6b shows the comparison of modelled total CO columns for FIR to the IASI retrievals for the same period. Overall EMAC captures the spatial CO pattern with overprediction of $1.0 \times 10^{18}$ molecules $\mathrm{cm^{-2}}$ in South America. Rosanka et al. (2020b) showed that EMAC predicts total methanol columns too high in this region, which is related to EMAC's tendency to simulate the Amazon basin too dry in

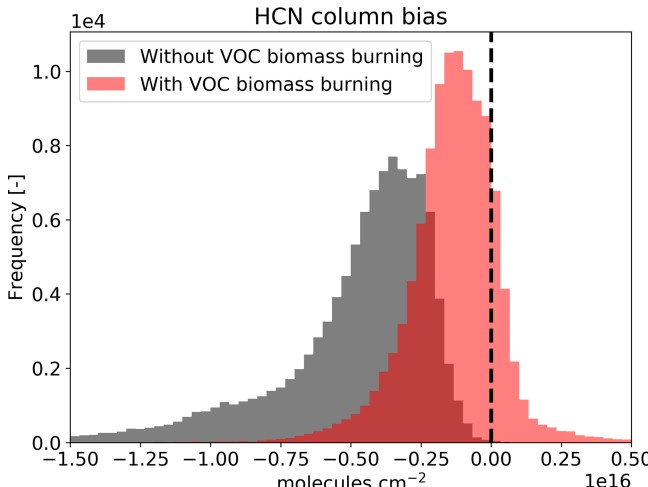

**Figure 5.** Global HCN column bias between EMAC simulations and IASI satellite data. The column bias is calculated based on monthly mean data during the Indonesian peatland fires in 2015.

the dry season (September–November) and consequently too hot (Hagemann and Stacke, 2015). This results in an overestimation of biogenic VOC emissions in South America. Since VOC degradation is the main atmospheric CO source, their overprediction explains EMAC's high bias for CO total columns in this region and its outflows.

As seen in Fig. 6b and c, EMAC constantly underestimate total CO columns over Indonesia during the main peatland fire period. However, overall the model bias stays within a factor of 2 (dashed lines in Fig. 6c). This underprediction can be explained by the emission factors used by EMAC. Stockwell et al. (2016) and Smith et al. (2018) report CO emission factors ranging from 216 to 314 g kg$^{-1}$ obtained from observation in Indonesia and Malaysia during the Indonesian peatland fires of 2015. In addition, in the recent assessment of Andreae (2019) a mean emission factor of 260 g kg$^{-1}$ for peatland is reported across multiple studies, which is higher than the emission factor used by EMAC (182 g kg$^{-1}$, Akagi et al., 2011).

From this analysis we conclude that even though EMAC does not reproduce HCN and CO columns perfectly, the Indonesian fires are reasonably well represented, especially when considering the exceptional strength of the 2015 Indonesian fires (for further discussion see Appendix A and Fig. A3). This also holds true considering all global biomass burning emission events.

## 5 The impact of biomass burning on the troposphere

In the following subsections, the impact of the 2015 Indonesian peatland fires on the lower tropospheric composition is analysed. In addition, substantial differences compared to the other six biomass-burning-dominated regions

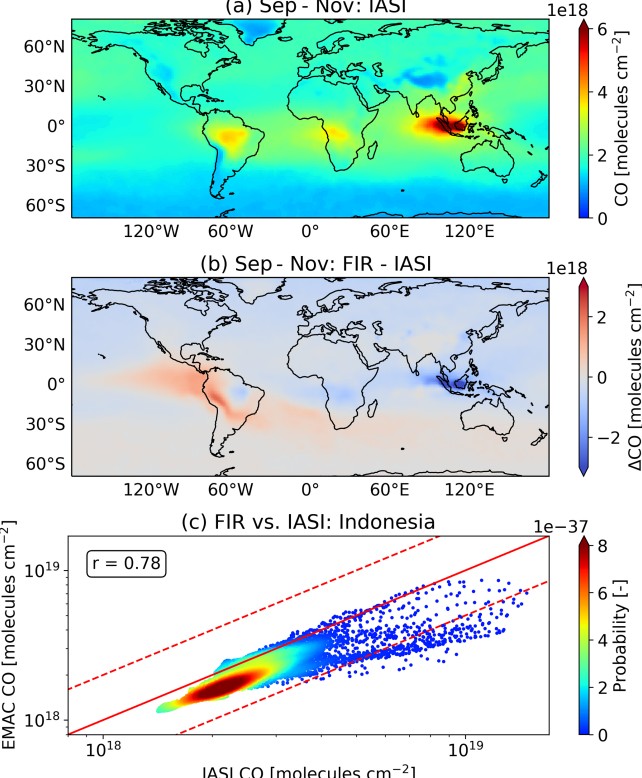

**Figure 6.** CO comparison between IASI and FIR. **(a)** Mean global observed IASI CO columns for September to November. **(b)** Mean global CO column comparison between FIR and IASI for September to November. **(c)** Scatter plot for direct comparison between FIR and IASI over Indonesia for September to November.

are discussed. All results are based on the simulations REF and FIR. Figure 7 depicts how the Indonesian peatland fires affect the atmospheric gas-phase composition. Table 3 provides an overview on the global and regional changes (between simulation REF and FIR) in the tropospheric burden of each species discussed in the following subsections. The regional changes reported in Table 3 are calculated for the respective main biomass burning season defined in Table 2.

### 5.1 Hydrocarbons

Many VOCs are characterised by short lifetimes resulting in highly-location-dependent changes within the troposphere. Globally, biomass burning emissions of VOCs significantly increase the atmospheric concentration of many hydrocarbons. In general, hydrocarbons can be separated into the aliphatic hydrocarbons and aromatic hydrocarbons. For both, direct emissions are the only atmospheric source.

Aliphatic hydrocarbons are further grouped into alkanes (only single covalent bonds), alkenes (containing at least one C−C double bond), and alkynes (containing at least one C−C triple bond). Ethane ($C_2H_6$) is globally the most abundant alkane and is impacted the most by biomass burning.

**Table 3.** Absolute (Abs.) and relative (Rel.) changes in the tropospheric burden for each region and each species discussed. Regional differences are calculated for the main biomass burning season (see Table 2), and the global changes are calculated for the complete year of 2015. The units for the absolute differences are explicitly given, whereas relative changes are always given in percent (%). Most radical burdens are presented in CEI moles (mol). The differences are calculated between simulation REF and FIR.

| Species | Unit* | Global Abs. | Global Rel. | ALA Abs. | ALA Rel. | CAF Abs. | CAF Rel. | CSA Abs. | CSA Rel. | NAS Abs. | NAS Rel. | NAU Abs. | NAU Rel. | SAF Abs. | SAF Rel. | SEA Abs. | SEA Rel. |
|---|---|---|---|---|---|---|---|---|---|---|---|---|---|---|---|---|---|
| **Aliphatic hydrocarbons** | | | | | | | | | | | | | | | | | |
| Ethane | Gg | 422.9 | 32.6 | 20.4 | 140.1 | 18.4 | 43.6 | 23.9 | 144.9 | 31.8 | 47.3 | 16.7 | 128.3 | 26.5 | 123.8 | 47.6 | 48.2 |
| Propane | Gg | 19.4 | 6.3 | 2.3 | 290.8 | 1.5 | 12.8 | 1.7 | 51.9 | 3.1 | 18.7 | 0.7 | 79.8 | 2.4 | 84.3 | 2.8 | 18.1 |
| n-Butane | Mg | 3727.4 | 2.4 | 470.4 | 242.7 | 256.1 | 6.0 | 336.3 | 16.9 | 671.3 | 8.6 | 86.3 | 33.2 | 324.9 | 34.9 | 779.6 | 9.4 |
| Ethylene | Gg | 13.5 | 12.5 | 2.0 | 103.5 | 2.9 | 59.9 | 3.6 | 9.2 | 2.2 | 46.3 | 0.8 | 26.4 | 4.5 | 81.5 | 8.3 | 85.3 |
| Propene | Mg | 3283.5 | 11.3 | 590.7 | 116.2 | 1021.5 | 83.4 | 863.5 | 5.4 | 560.7 | 54.3 | 279.5 | 34.5 | 1420.7 | 100.9 | 2304.0 | 83.6 |
| Isobutene | Mg | 107.1 | 20.2 | 36.2 | 7299.4 | 21.7 | 103.8 | 41.0 | 379.8 | 29.3 | 293.8 | 4.1 | 380.5 | 27.0 | 161.0 | 53.8 | 70.9 |
| Acetylene | Gg | 45.7 | 20.5 | 1.4 | 119.8 | 4.4 | 56.0 | 6.0 | 305.0 | 2.5 | 20.8 | 2.8 | 258.8 | 6.6 | 184.5 | 13.2 | 64.2 |
| **Aromatic hydrocarbons** | | | | | | | | | | | | | | | | | |
| Benzene | Gg | 38.8 | 27.3 | 4.4 | 1312.7 | 2.7 | 46.4 | 4.3 | 2.2 | 5.9 | 84.2 | 1.9 | 175.9 | 4.6 | 228.7 | 20.0 | 207.7 |
| Toluene | Mg | 6655.3 | 15.3 | 802.6 | 1308.7 | 595.5 | 62.8 | 1007.7 | 85.0 | 968.2 | 62.0 | 198.2 | 198.2 | 950.0 | 199.0 | 366.8 | 366.8 |
| Styrene | Mg | 56.2 | 29.8 | 42.8 | 11375.3 | 8.0 | 270.4 | 10.2 | 151.1 | 31.7 | 627.7 | 2.1 | 181.9 | 13.2 | 582.3 | 10.0 | 35.6 |
| Ethylbenzene | Mg | 2431.9 | 65.9 | 774.2 | 27408.4 | 421.5 | 958.0 | 351.8 | 470.8 | 820.3 | 592.5 | 139.2 | 1377.7 | 731.0 | 3203.8 | 237.3 | 75.4 |
| **OVOCs** | | | | | | | | | | | | | | | | | |
| Formaldehyde | Gg | 25.4 | 2.2 | 3.6 | 35.1 | 5.4 | 11.7 | 5.3 | 2.2 | 3.4 | 12.6 | 1.6 | 3.9 | 8.4 | 17.9 | 15.9 | 80.7 |
| Acetaldehyde | Gg | 15.5 | 11.4 | 1.0 | 79.0 | 2.6 | 47.7 | 4.2 | 6.9 | 6.9 | 31.2 | 0.9 | 22.5 | 3.7 | 74.3 | 8.8 | 80.7 |
| Glycolaldehyde | Gg | 21.3 | 8.0 | 1.7 | 65.6 | 3.7 | 27.8 | 6.9 | 5.1 | 5.1 | 36.5 | 1.2 | 9.5 | 5.8 | 34.7 | 17.2 | 67.1 |
| Methanol | Gg | 223.3 | 7.9 | 11.7 | 31.6 | 17.0 | 16.8 | 31.5 | 6.6 | 16.2 | 16.2 | 14.4 | 17.5 | 27.4 | 28.5 | 112.3 | 60.7 |
| Glyoxal | Mg | 3872.3 | 9.3 | 126.2 | 388.7 | 591.5 | 24.5 | 397.3 | 3.9 | 591.5 | 38.6 | 240.1 | 12.7 | 1048.3 | 36.7 | 3186.1 | 62.2 |
| Methyl glyoxal | Mg | 2481.7 | 1.3 | 17.4 | 208.1 | 206.5 | 4.1 | 212.4 | 0.2 | 144.4 | 6.9 | 1.3 | 12.7 | 743.4 | 5.8 | 2195.8 | 10.2 |
| 2,3-Butanedione | Mg | 487.1 | 205.5 | 0.4 | 180.5 | 319.7 | 812.6 | 0.2 | 8060.4 | 1.0 | 21.0 | 6.4 | 960.4 | 94.2 | 2588.8 | 807.0 | 304.2 |
| Phenol | Mg | 1167.8 | 105.7 | 171.8 | 4282.2 | 275.7 | 316.4 | 155.7 | 1305.1 | 323.5 | 323.5 | 63.9 | 1353.8 | 277.1 | 1339.7 | 1226.3 | 49.9 |
| Benzaldehyde | Mg | 282.1 | 14.6 | 102.9 | 4472.0 | 48.6 | 196.9 | 83.2 | 92.4 | 92.4 | 221.3 | 19.4 | 284.9 | 108.2 | 574.7 | 125.2 | 49.9 |
| CO | Gg | 8341.1 | 2.4 | 153.3 | 3.9 | 446.4 | 2.4 | 307.5 | 2.8 | 503.2 | 2.6 | 503.2 | 6.2 | 273.4 | 3.4 | 1908.0 | 6.5 |
| **Acids** | | | | | | | | | | | | | | | | | |
| Formic acid | Gg | 32.9 | 4.9 | 1.5 | 24.0 | 7.0 | 25.2 | 4.7 | 15.3 | 2.1 | 12.9 | 3.2 | 13.0 | 11.4 | 49.9 | 11.3 | 20.7 |
| Acetic acid | Gg | 119.3 | 23.3 | 8.7 | 441.7 | 29.0 | 124.3 | 9.8 | 15.3 | 9.8 | 128.5 | 10.4 | 45.3 | 48.7 | 238.9 | 37.7 | 117.4 |
| **Oxidants** | | | | | | | | | | | | | | | | | |
| O₃ | Gg | 1115.3 | 0.3 | 92.0 | 1.9 | 61.6 | 0.7 | -28.3 | -0.3 | 104.7 | 0.7 | 2.0 | 0.0 | 60.4 | 0.8 | 83.2 | 0.4 |
| OH | kmol | -240.3 | -1.7 | -6.4 | -4.9 | -8.5 | -2.4 | -15.9 | -4.7 | -10.8 | -3.4 | -21.6 | -6.8 | -10.3 | -3.5 | -58.2 | -4.5 |
| HO₂ | kmol | 3537.2 | 0.4 | 354.6 | 3.7 | 410.0 | 1.7 | 367.6 | 0.8 | 409.6 | 1.8 | 157.7 | 0.7 | 533.4 | 2.8 | 945.4 | 1.3 |
| NO | Mmol | -82.3 | -2.8 | -38.4 | -52.7 | -5.9 | -6.5 | -5.7 | -6.6 | -26.0 | -26.6 | -6.8 | -10.6 | -6.4 | -7.5 | -22.0 | -7.7 |
| NO₂ | Mmol | -178.5 | -2.2 | -73.5 | -36.7 | -16.3 | -4.7 | -14.2 | -4.1 | -60.1 | -19.3 | -13.0 | -6.5 | -18.8 | -5.5 | -70.5 | -8.3 |
| NO₃ | kmol | 6926.5 | 5.3 | 151.3 | 35.5 | 385.0 | 6.0 | 193.8 | 6.6 | 407.2 | 15.2 | 497.8 | 16.9 | 432.3 | 9.3 | 2368.5 | 15.5 |

\* Unit for absolute values only. Relative values given in percent (%).

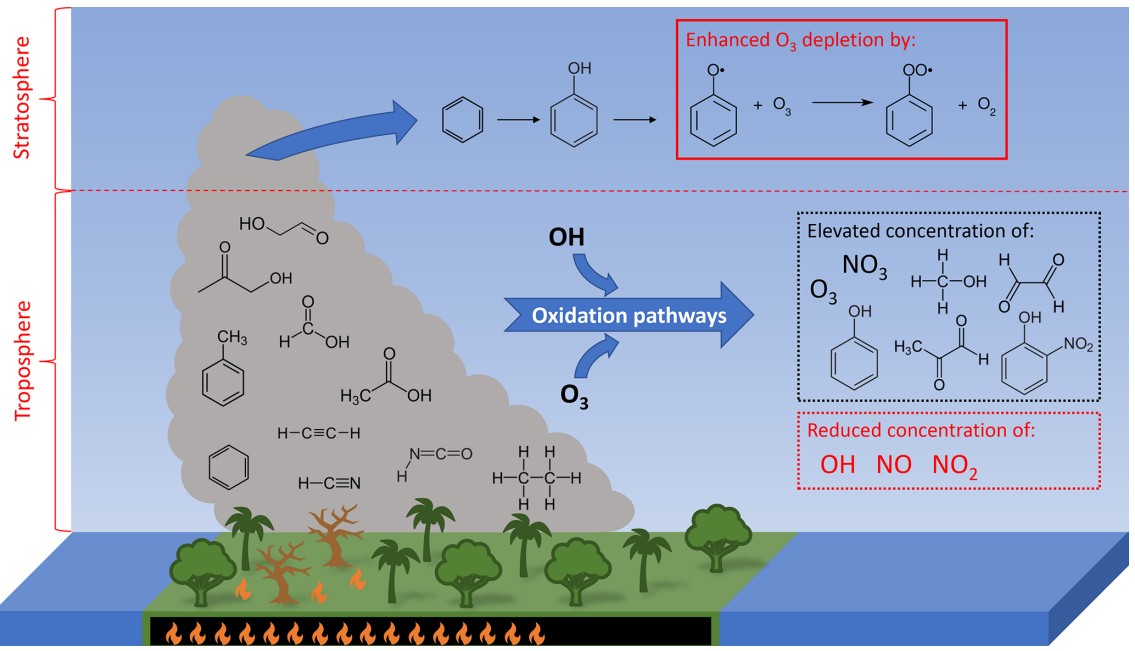

**Figure 7.** Illustration of the impact of VOC emissions from the Indonesian peatland fires on the atmospheric composition.

Its global burden is increased by 32.6 %, whereas the burden of less abundant alkanes like propane ($C_3H_8$) and *n*-butane ($C_4H_{10}$) only increases by 6.3 % and 2.4 %, respectively. Overall, the global change in the burden of alkenes is lower than that of alkanes. Here, ethylene ($C_2H_4$) has the highest absolute change of 13.5 Gg (12.5 %) followed by propene ($C_3H_6$) with 3.3 Gg (11.3 %). Even tough its abundance is the lowest, the highest global relative change of 20.2 % is predicted for isobutene ($C_4H_8$). In addition, EMAC predicts an increase of 20.5 % due to biomass burning emissions for the alkyne acetylene ($C_2H_2$). In general, the highest absolute change is predicted for SEA, except for propane since its burden increase is 0.3 Gg higher in NAS than in SEA. In both cases, the relative change is very similar. For many aliphatic hydrocarbons, the lowest absolute changes are predicted in ALA. However, due to the generally low background concentrations in this area, the relative changes are the highest, making biomass burning in this region the major source of these hydrocarbons.

The two most abundant aromatic hydrocarbons, benzene ($C_6H_6$) and toluene ($C_6H_5CH_3$), are strongly emitted by biomass burning events. In the FIR simulation, the tropospheric burden of benzene increases by 27.3 %. Toluene has a slightly lower increase of only 15.3 %. A higher relative impact is predicted for less abundant aromatics like ethylbenzene ($C_8H_{10}$) and styrene ($C_8H_8$). Here, the global burden changes by 65.9 % and 29.8 %, respectively. As it is for the aliphatic hydrocarbons, the highest absolute changes for benzene and toluene are predicted in SEA during the Indonesian peatland fires. Opposite to this, EMAC predicts the lowest change in SEA for ethylbenzene and styrene, which is re-lated to the fact that EMAC uses significant lower emissions for both aromatic hydrocarbons for peatland when compared to the recent values reported by Andreae (2019).

## 5.2 Oxygenated organics

The degradation of aliphatic and aromatic hydrocarbons leads to the formation of oxygenated organic compounds. Additionally, they are emitted by biomass burning such as the Indonesian peatland fires. Globally, biomass burning has only a little impact on formaldehyde (HCHO), the simplest aldehyde (R−CHO). However, regional changes are predicted to be higher and range from 2.2 % to 35.1 %. The global and regional changes are higher for more complex aldehydes. The global burden of acetaldehyde ($CH_3CHO$) and glycolaldehyde ($HOCH_2CHO$) increases by 11.4 % and 8.0 %, respectively. In all cases, the highest absolute and rel-ative change is predicted in SEA. The two $\alpha$-dicarbonyls gly-oxal (OCHCHO) and methyl glyoxal ($CH_3C(O)CHO$) are primarily produced from VOC oxidation. Their global bur-den increases by 9.3 % and 1.3 %, respectively. Again, the highest absolute changes are predicted in SEA. However, the highest relative change occurs in ALA due to generally low background VOC concentrations. Globally, methanol ($CH_3OH$) increases by 7.9 % when biomass burning VOC emissions are taken into account. Here, the Indonesian peat-land fires contribute by far the most. A significantly higher impact is predicted for 2,3-butanedione (($CH_3CO$)$_2$). Its global burden is tripled due to biomass burning, and the ab-solute changes predicted regionally are the highest in NAS and SEA.

In the atmosphere, organic acids are mainly produced from the photo-oxidation of biogenic and anthropogenic VOCs but may also be emitted from biomass burning. Formic acid (HCOOH) is slightly impacted by biomass burning VOC emissions and globally increases by 4.9 % with the highest changes in SEA and Africa (CAF and SAF). The acid impacted the most by biomass burning is acetic acid (CH$_3$COOH), which globally gains 23.3 % with the highest changes in SEA, CAF, CSA, and SAF. Interestingly, the high increase predicted in CSA only leads to a low relative rise. This is due to generally high background concentrations in this region from high biogenic VOC emissions.

The largest change in oxygenated aromatics is predicted for phenol (C$_6$H$_5$OH), whose tropospheric burden is more than doubled and increases to 2.3 Gg. Even though phenol is directly emitted by biomass burning, the overall high aromatic emissions lead to an enhanced chemical production of phenol from benzene oxidation. The highest absolute change is observed in SEA. However, due to low aromatic background concentrations, the relative increase is higher in ALA, CSA, and NAU. The increase in benzaldehyde (C$_6$H$_5$CHO) is significantly lower (globally by 14.6 %) with similar absolute changes in ALA, CAF, NAS, SAF, and SEA.

The oxidation of VOCs leads to the formation of CO (see Sect. 4.2). Overall, the VOC emissions from biomass burning only result in a global CO increase of 2.4 %, with regional changes between 2.4 % and 6.5 %.

## 5.3 Nitrogen-containing compounds

Besides looking at HCN, we also analysed the impact of the Indonesian peatland fires on two nitrogen-bearing compounds that are toxic for humans (isocyanic acid) and for vegetation (nitrophenols). Isocyanic acid (HNCO) is known to be a toxic constituent of biomass burning emissions. It is linked to protein carbamylation, which causes adverse health effects such as rheumatoid arthritis, cardiovascular diseases, and cataracts (Wang et al., 2007; Roberts et al., 2011; Leslie et al., 2019). It is expected that the protein carbamylation potentially starts if humans are exposed to ambient concentrations above 1 ppb (Roberts et al., 2011). Rosanka et al. (2020d) already reported that HNCO concentrations are high in regions characterised by strong biomass burning events. Globally, similar high concentrations are predicted in this study. However, we predict higher concentrations in Indonesia than Rosanka et al. (2020d), who reported that ambient HNCO conditions of 1 ppb are exceeded for less than 30 d in Indonesia in 2011. The year 2011 is known to have low biomass burning emissions in this region (van der Werf et al., 2017). Figure 8 shows the number of days in which this threshold is exceeded during the 2015 Indonesian peatland fires between August and October. Here, 1 ppb of HNCO is regularly exceeded, and some regions are affected during the complete fire period. This causes potentially severe health effects for the population of Indonesia, which is the world's fourth highest (United Nations, 2019).

In the atmosphere, nitrophenols are mainly formed from the oxidation of the aromatic compounds benzene, toluene, phenols, and cresols (Nojima et al., 1975; Atkinson et al., 1980; Grosjean, 1984), of which the first three are emitted by biomass burning (see Sect. 5.1 and 5.2). Without biomass burning emissions of aromatics, the modelled nitrophenol concentrations are only high in regions with high anthropogenic activities (Fig. 9a). When biomass burning emissions of benzene, toluene, and phenols are included, nitrophenol concentrations significantly increase in areas affected by biomass burning. The strongest changes occur in SEA, CAF, and SAF (Fig. 9b). Many biomass burning regions frequently exceed nitrophenol thresholds that are determined for regions where anthropogenic aromatic emissions dominate. On a global scale, biomass burning becomes the main source of nitrophenols. Nitrophenols are known to have a high phytotoxic activity that is prolonged given their photochemical stability (Grosjean, 1991). Rippen et al. (1987) and Natangelo et al. (1999) suggested that nitrophenols could have contributed to the forest decline in northern and central Europe in the 1980s but also in other parts of the world. Therefore, the overall increase in nitrophenols in biomass burning areas is a potential danger for plants in these regions where plants are already under stressed conditions due to the biomass burning itself. At the same time, nitrophenols are known to absorb solar radiation (Hems and Abbatt, 2018) and therefore enhance hazy conditions in those areas (Lee et al., 2017), contributing to increased morbidity and mortality (Crippa et al., 2016).

## 5.4 Radicals

In general, organic molecules react with OH by either H abstraction or addition to double bonds, making OH the most important daytime VOC oxidant. Figure 10a gives the mean tropospheric surface OH concentration in 2015, and Fig. 10b presents the changes due to biomass burning VOC emissions. OH concentrations are significantly reduced in most regions with frequent biomass burning events. This reduction is caused by the direct reaction of OH with VOCs and the enhanced formation of CO from VOC degradation. The reduction in OH is not uniformly distributed and depends on the local chemical regime. In Indonesia, the high VOC emissions lead to the highest absolute and relative OH reduction. The enhanced oxidation of VOCs by OH leads to an overall increase in HO$_2$. In ALA and NAS, the most northern areas of interest, the absolute change in OH is low (see Table 3). Within the biomass burning plume, the enhanced HO$_2$ concentrations react with NO, producing OH and compensating for the OH reduction by VOC degradation, resulting in a regional surface OH increase. Still, outside the biomass burning plume, an overall decrease in OH is predicted in ALA and NAS. Here, VOCs from biomass burning become the highest

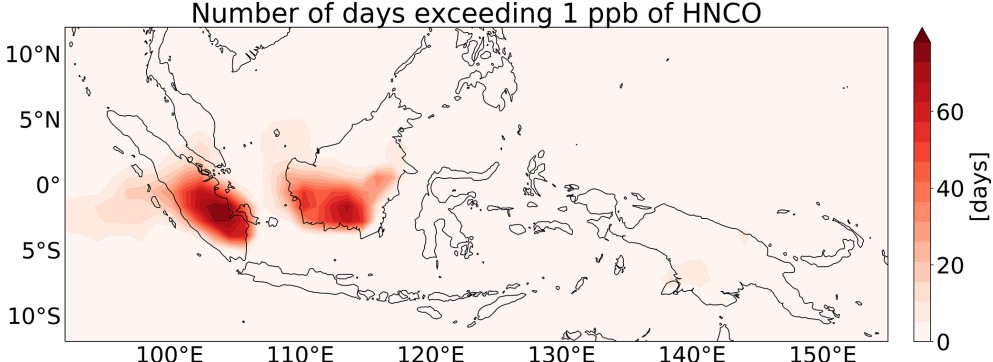

**Figure 8.** Number of days in which ambient concentrations of 1 ppb of HNCO are exceeded during the Indonesian peatland fires in 2015 between August and October.

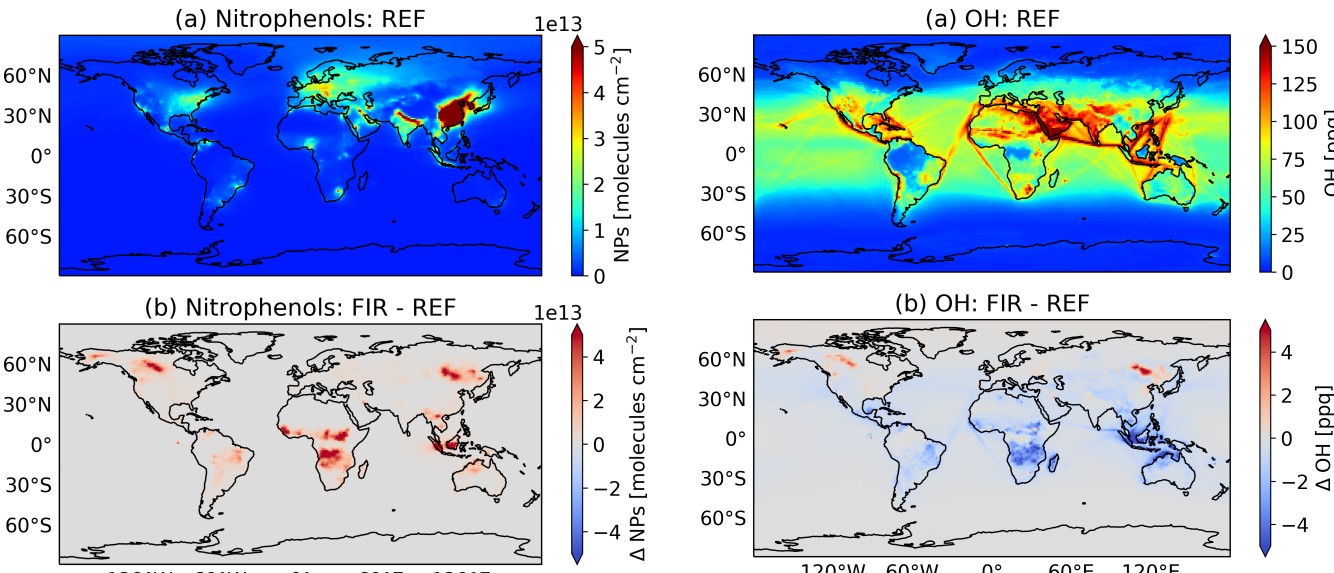

**Figure 9.** (a) Yearly mean tropospheric nitrophenol (NPs) column without biomass burning VOC emissions. (b) Changes in the yearly mean tropospheric nitrophenol (NPs) due to VOC biomass burning.

**Figure 10.** (a) Yearly mean surface OH concentration without biomass burning VOC emissions. (b) Changes in the yearly mean surface OH concentration due to VOC biomass burning.

OH sink, resulting in strong relative changes in OH reactivity. In general, OH reactivity is the highest in the Amazon basin ($100\,s^{-1}$) and the lowest in Antarctica ($0.5\,s^{-1}$). The additional VOC emissions in Indonesia result in a significant increase of about 50 % in the OH reactivity, which is similar to the increases predicted in ALA and NAS.

Figure 11a and b show the mean surface $NO_x$ concentrations and the changes induced by the VOC biomass burning emissions, respectively. The additional VOC emissions significantly reduce the regional concentrations in tropospheric $NO_x$. In SEA, the absolute changes are large but small relatively (about 8 %), whereas the highest absolute and relative $NO_x$ changes are predicted in ALA. These reductions are caused by enhanced reactions of $RO_2$ with $NO_x$, resulting in an increased formation of $NO_x$ reservoir species (i.e. alkyl and acyl peroxy nitrates) and nitrogen-containing aromatics (e.g. nitrophenols).

$NO_3$ is the most important nighttime oxidant, which is globally increased by about 5 % when the biomass burning emissions of VOCs are included (see Table 3). On the one hand, the formation of $NO_3$ is enhanced by aromatic $RO_2$ reacting with $NO_2$, but on the other hand the loss of $NO_3$ by reactions with $RO_2$ and aldehydes is increased. In the two northern regions (ALA and NAS), the elevated $O_3$ and regionally increased $NO_2$ concentrations induce an enhanced formation from inorganic reactions, resulting in an additional rise of $NO_3$. The absolute increase in $NO_3$ is high in SEA, especially in Indonesia. Here, the particularly large increase in phenols results in enhanced concentrations of phenyl peroxy radicals ($C_6H_5O_2$), which form $NO_3$ when reacting with

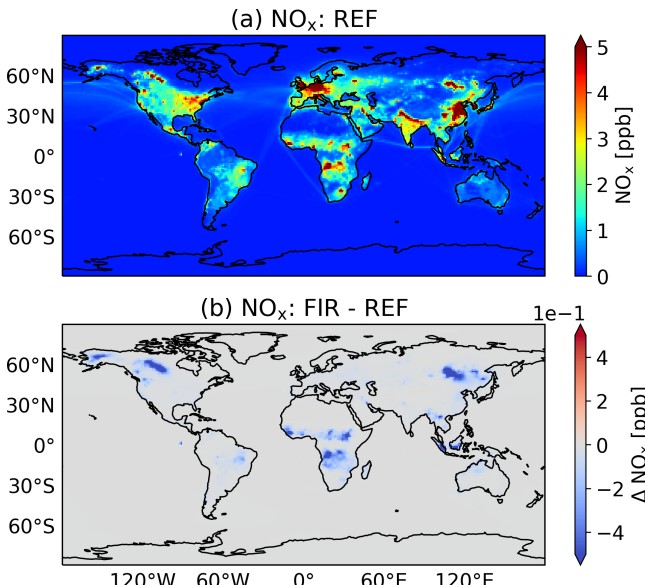

**Figure 11. (a)** Yearly mean surface $NO_x$ concentration without biomass burning VOC emissions. **(b)** Changes in the yearly mean surface $NO_x$ concentration due to VOC biomass burning.

$NO_2$ following Jagiella and Zabel (2007):

$$ \text{(structure)} + NO_2 \longrightarrow \text{(structure)} + NO_3 \tag{R1} $$

Taraborrelli et al. (2021) recently studied the importance of aromatics on the atmospheric composition on a global scale. They also demonstrated the importance of this reaction but, in opposition to our findings, predicted a reduction of $NO_3$ in Indonesia. Taraborrelli et al. (2021) used a different resolution and analysed 2010, a year with little biomass burning emissions in Indonesia (van der Werf et al., 2017), reducing the importance of this production channel.

## 5.5 Ozone

The perturbed $NO_x$–$HO_x$ relation consequently leads to changes in tropospheric $O_3$. Overall, EMAC predicts an enhanced formation of $O_3$ when the biomass burning emissions are included (see Table 3). The increase in $HO_2$ leads to an enhanced chemical $O_3$ production by reacting with NO. Due to high $NO_x$ emissions from biomass burning, the $O_3$ production is to a large extent VOC-limited. In the two northern regions, the background VOC concentrations are low, resulting in the highest relative changes of more than 10 %. Based on the long-term simulations (REF_LONG and FIR_LONG) we find that the largest changes are predicted in the NH high latitudes in 2003, a year with intense biomass burning in boreal Asia (van der Werf et al., 2017). However, compared to the averaged tropospheric background $O_3$ concentrations, these

changes are negligible on a global scale. As described in Sect. 4, most VOC emissions from Indonesia are transported towards the Indian Ocean. Therefore, $O_3$ is predicted to increase in Sumatra and west of it. Interestingly, away from biomass burning emissions in Kalimantan and in east Indonesia, $O_3$ concentrations are slightly reduced, even though the chemical $O_3$ production still increases in this area. The particularly strong emissions of aromatics lead to enhanced concentrations of phenoxy radicals ($C_6H_5O$), which directly destroy $O_3$ (Tao and Li, 1999) in lower $NO_x$ regions:

$$ \text{(structure)} + O_3 \longrightarrow \text{(structure)} + O_2 \tag{R2} $$

This $O_3$ sink increases by 780 %, resulting in a net loss of $O_3$ in these areas. Globally, this $O_3$ destruction channel increases from 144.9 to 200.1 $Tg\,a^{-1}$ in the troposphere. Also, Taraborrelli et al. (2021) reported a similar strength of this destruction channel of about 200 $Tg\,a^{-1}$. Therefore, biomass burning emissions regionally control the importance of this destruction channel.

## 6 The influence of Indonesian peatland fires on the UTLS

As illustrated in Fig. 7, some of the biomass burning VOC emissions from Indonesia are quickly transported by the ASMA and the general tropical updraught into the UTLS (see Sect. 1 and Vogel et al., 2015). In the following, we define the lower stratosphere between 147–32 hPa (about 13–24 km) above 30° in latitudinal direction and between 100–32 hPa (about 17–24 km) below 30° latitude. In Table 4, the lower stratospheric burden in November for each discussed species is presented, including the changes induced by biomass burning and the contribution from the Indonesian peatland fires.

### 6.1 Hydrocarbons

Even though their atmospheric lifetime is generally short, the upward transport in the tropics leads to an increase in hydrocarbons in the lower stratosphere due to biomass burning. Similar to the changes in the troposphere, the aliphatic hydrocarbon with the highest absolute change is ethane. The lower stratospheric burden in November increases by about 41 % to 43.56 Gg. The lower stratospheric burden of other aliphatic hydrocarbons like propane and $n$-butane changes by around 30 %. The lower stratospheric burden of benzene is tripled, whereas toluene is doubled, which is consistent with the difference between their chemical lifetimes. The contribution of the Indonesian peatland fires for most hydrocarbons ranges between 69 % and 87 %, except ethane. We expect that additional non-Indonesian fires from SEA contribute the rest,

**Table 4.** Stratospheric burden in November 2015 and changes induced by VOC biomass burning emissions. In addition, the relative difference (Rel.) and the Indonesian contribution (Indo. contr.) are shown. The latter is calculated based on the difference of FIR and $FIR_{NOINDO}$.

| Species | Unit | REF | $\Delta$FIR | Rel. [%] | Indo. contr. [%] |
|---------|------|-----|-------------|----------|------------------|
| Ethane | Gg | 30.8 | 43.6 | 41.6 | 24.1 |
| Propane | Mg | 666.3 | 855.9 | 28.5 | 69.1 |
| *n*-Butane | Mg | 116.2 | 152.2 | 31.0 | 80.0 |
| Benzene | Mg | 242.1 | 724.2 | 199.2 | 87.2 |
| Methanol | Gg | 17.1 | 22.9 | 34.0 | 76.0 |
| Glyoxal | Mg | 9.8 | 14.7 | 48.9 | 69.4 |
| Phenol | Kg | 629.8 | 1685.6 | 167.7 | 85.8 |
| Acetic acid | Mg | 289.4 | 473.6 | 63.7 | 72.6 |
| HCN | Gg | 16.7 | 53.5 | 220.4 | 62.6 |
| Nitrophenols | Mg | 14.7 | 23.6 | 60.2 | 69.5 |
| OH | Mmol | 3.8 | -0.1 | -3.2 | 70.2 |
| NO | Gmol | 1.9 | -0.1 | -6.1 | 67.7 |
| $NO_2$ | Gmol | 3.6 | -0.2 | -4.2 | 66.2 |
| $NO_3$ | Mmol | 10.3 | 1.0 | 9.5 | 74.7 |
| $HO_2$ | Mmol | 24.1 | 0.6 | 2.4 | 63.9 |

since the Indonesian peatland fires contribute about 76 % to the total biomass burning emissions from SEA in 2015. In the case of ethane, the contribution from the Indonesian peatland fires is only about 24 %. Compared to other hydrocarbons, ethane has an atmospheric lifetime of about 2 months (Hodnebrog et al., 2018). Thus, we expect that its long lifetime allows ethane emitted from other biomass burning regions to be transported into the lower stratosphere. At the same time, the recent biomass burning inventory by Andreae (2019) indicates that EMAC underestimates ethane emissions from the Indonesian peatland fires by a factor of 3.

## 6.2 Oxygenated organics

In addition to the upward transport of the directly emitted OVOCs, the elevated hydrocarbon concentrations also form OVOCs in the lower stratosphere. In the lower stratosphere, methanol is one of the most abundant OVOCs, and its burden increases by 34 %. EMAC predicts a higher relative increase for less abundant OVOCs like glyoxal (about 49 %) and acetic acid (about 64 %). The high increase in benzene, due to the strong aromatic emissions from the Indonesian peatland fires (see Sect. 3), results in the particularly large production of phenol. Here, the lower stratospheric burden increases by about 167 %. The contribution from the Indonesian peatland fires to the lower stratospheric burden of all OVOCs is in a similar range as for the hydrocarbons, namely from about 72 % to 86 %.

## 6.3 Nitrogen-containing compounds

The Indonesian peatland fires resulted in substantial HCN emissions, as seen in Fig. 4 and as discussed in Sect. 4.1, which results in a strong increase in HCN in the lower stratosphere. Here, EMAC predicts an increase of more than 36 Gg

(about 220 %). Sheese et al. (2017) report that the highest increase on record in lower stratospheric HCN was observed by the Atmospheric Chemistry Experiment Fourier transform spectrometer (ACE-FTS) instrument on the SCISAT satellite following the 2015 Indonesian peatland fires. Therefore, based on the long-term simulations (simulation $REF_{LONG}$ and $FIR_{LONG}$), EMAC reproduces their findings. The elevated lower stratospheric benzene and toluene concentrations lead to an increase in lower stratospheric nitrophenol concentrations of about 60 %.

## 6.4 Radicals

The oxidation of VOCs transported into the lower stratosphere influences the lower stratospheric oxidation capacity. Overall, the lower stratospheric OH burden is reduced by about 3 %, whereas the burden of $HO_2$ increases by 2.4 % (see Table 4). The enhanced formation of $NO_x$ reservoir species results in a 6 % and 4.2 % reduction of NO and $NO_2$, respectively. At the same time, the enhanced reactions of $NO_2$ with aromatic $RO_2$ results in an increase in $NO_3$ of more than 9 %. Figure 12 provides the mean longitudinal relative change in lower stratospheric OH, $HO_2$, $NO_x$, and $NO_3$ between 2001 and 2016 based on the long-term simulations (simulation $REF_{LONG}$ and $FIR_{LONG}$). After each Indonesian peatland fire period, the lower stratospheric oxidants are influenced. With decreasing lower stratospheric VOC concentrations over time, the influence on the oxidants vanishes in the second half of the following year. Particularly strong influences are observed during El Niño periods, caused by enhanced VOC emissions from peatland fires. For example, intense fires in 2006 and 2015 led to a significant change in lower stratospheric oxidants in early 2007 and 2016, respectively. In 2010, almost no fires occurred in Indonesia (van der Werf et al., 2017), resulting in only a little change in oxidant

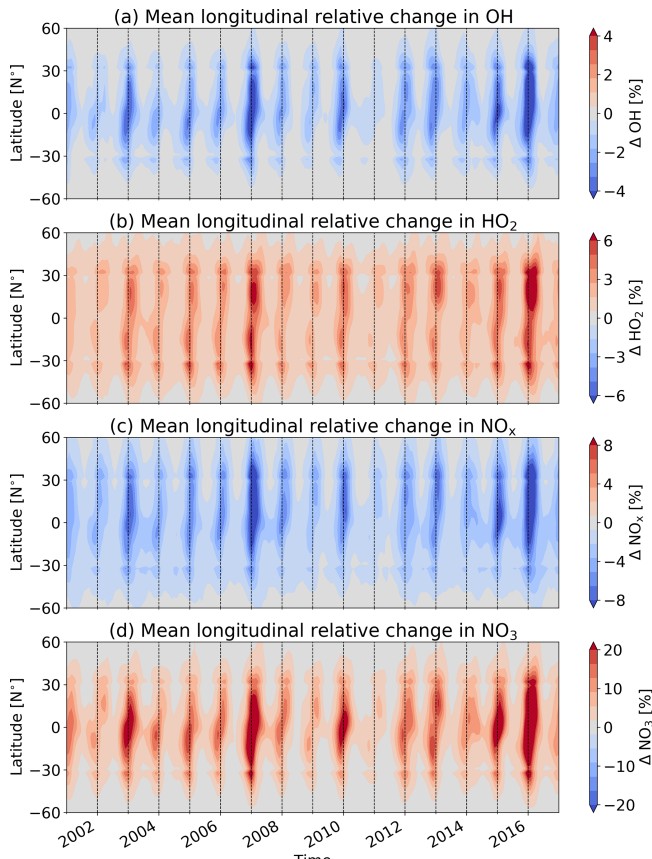

**Figure 12.** The mean longitudinal relative change in lower stratospheric (**a**) OH, (**b**) $HO_2$, (**c**) $NO_x$, and (**d**) $NO_3$ between 2001 and 2016. Results are based on both long-term simulations (simulation $REF_{LONG}$ and $FIR_{LONG}$). The lower stratosphere is defined between 147–32 hPa (about 13–24 km) above 30° in latitudinal direction and between 100–32 hPa (about 17–24 km) below 30° latitude.

concentrations in 2011. Even in non-El Niño years, EMAC predicts changes in lower stratospheric radicals due to VOC biomass burning emissions. We expect that these originate from other biomass burning events in SEA (see Sect. 6.1). In each year, the highest influence is predicted for $NO_3$. Following the intense fires of 2002, 2006, 2009, 2014, and 2015 the zonal mean $NO_3$ concentration changes by more than 20 % in the tropical and subtropical lower stratosphere at the end of the same year and at the beginning of the next year. Thus, our findings indicate that VOC emissions from the Indonesian peatland fires quickly transported into the lower stratosphere become a major source of lower stratospheric $NO_3$.

## 6.5 Ozone

The elevated phenol and consequently phenoxy radicals in the UTLS influence the importance of the $O_3$ loss due to reaction with phenoxy radicals (Reaction R2). In the upper tropical troposphere this loss process contributes significantly to the total chemical $O_3$ loss. Figure 13a shows the zonal mean

in the relative contribution of this $O_3$ loss pathway to the total chemical $O_3$ loss without VOC biomass burning emissions in November 2015. Especially in the upper northern tropical troposphere, this loss process contributes up to 40 % to the total chemical $O_3$ loss. Following the benzene emissions from biomass burning, elevated phenoxy radicals in the UTLS double the contribution of this $O_3$ loss processes in the upper southern tropical troposphere (i.e. an increase of more than 20 %) and increase the contribution in the upper northern tropical troposphere by about 10 % in November 2015 (see Fig. 13b). A similar impact can be observed following other intense Indonesian biomass burning seasons. Figure 13c shows the zonal mean change in the relative contribution of this $O_3$ loss pathway in the UTLS between 2001 and 2016. Here, we define the UTLS from 250 to 50 hPa above the tropopause calculated by EMAC. Especially after the intense Indonesian peatland fires during strong El Niño periods, a change in the upper southern tropical UTLS of more than 10 % is predicted at the end of each year. In all years, the increase in the upper northern tropical UTLS is lower. Figure 14 shows the zonal mean relative change in the phenoxy radical $O_3$ loss pathway due to VOC biomass burning in April 2016. Following the increase in benzene in the lower stratosphere (Sect. 6.1), EMAC predicts an increase in this $O_3$ loss process by more than 400 %, which propagates into the upper tropical lower stratosphere. These findings suggest that the frequent re-occurrence of strong Indonesian peatland fires could contribute to the variability in lower stratospheric $O_3$ which is observed by remote sensing measurements (Kyrölä et al., 2013; Nair et al., 2015; Vigouroux et al., 2015; Chipperfield et al., 2018).

In the UTLS, aviation is the only direct anthropogenic activity and contributes about 3.5 % to the total anthropogenic climate change (Lee et al., 2021). Here, aviation $NO_x$ emissions lead to a formation of $O_3$ and a depletion of methane ($CH_4$). Recently, Rosanka et al. (2020a) showed that the enhancement in $O_3$ is limited by the background concentrations of $NO_x$ and $HO_x$. If enough $HO_x$ is available, a lower background $NO_x$ concentration results in a higher $O_3$ gain. In general, low background $HO_x$ concentrations limit the $O_3$ gain in winter. In our study, we find (not shown) that in the North Atlantic flight sector (between 400–100 hPa), the $NO_x$ burden is reduced due to VOC emissions from SEA fires by about 6 %, with regional changes of more than 20 % in 2015. At the same time, $HO_x$ increases regionally by 10 %. Even though $NO_x$ emissions from the frequently occurring Indonesian peatland fires are expected to result in an increase in UTLS $NO_x$, substantial VOC emissions from the same fires potentially compensate for the impact of the $NO_x$ increase and favour the formation of $O_3$ from aviation activities. In the simulation setup used, EMAC neglects VOC emissions reported from aviation activity (e.g. Wilkerson et al., 2010). Our findings indicate that the direct emissions of benzene, toluene, and phenol in the UTLS potentially enhance the loss

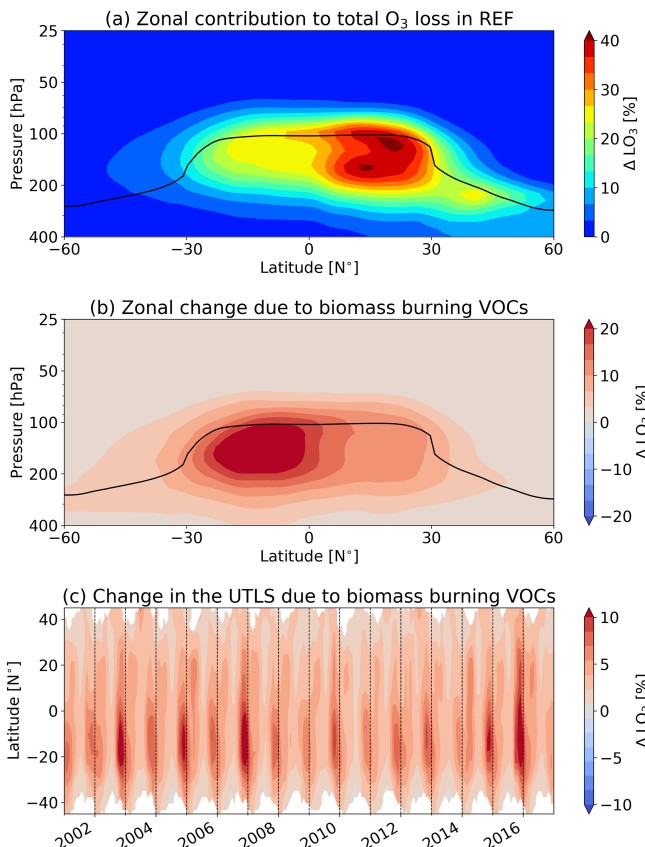

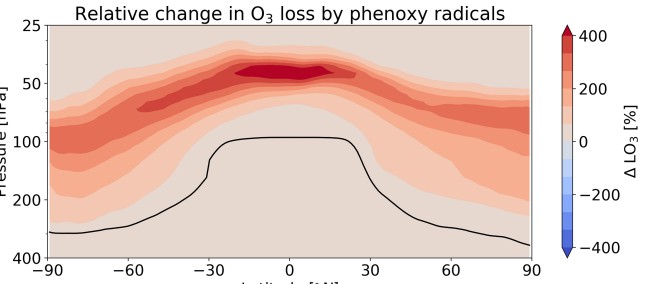

**Figure 14.** Zonal mean relative change in the destruction of $O_3$ by phenoxy radicals (Reaction R2) due to VOC biomass burning emissions (FIR vs. REF) in April 2016. The tropopause pressure level is indicated in black.

**Figure 13.** Zonal mean contribution of the destruction of $O_3$ by phenoxy radicals (Reaction R2) to the total chemical $O_3$ loss. **(a)** Relative contribution if no VOC biomass burning emissions are taken into account (REF) in November 2015. **(b)** Changes in the relative contribution due to VOC biomass burning emissions (FIR) in November 2015. **(c)** Changes in the relative contribution due to VOC biomass burning emissions in the UTLS. Here, we define the UTLS from 250 to 50 hPa above the tropopause calculated by EMAC. The tropopause pressure level in panels **(a)** and **(b)** is indicated in black.

of $O_3$ by phenoxy radicals and consequently affect the impact of aviation on $O_3$ in that region.

## 7 The influence of in-cloud OVOC oxidation

Recently, Rosanka et al. (2020b) showed that the in-cloud oxidation of OVOCs significantly influences tropospheric VOCs and oxidants. By using JAMOC (Rosanka et al., 2020c, b), we investigate the importance of in-cloud OVOC oxidation during the Indonesian peatland fires on the troposphere and lower stratosphere (simulations REF$_{\text{JAMOC}}$ and FIR$_{\text{JAMOC}}$). In order to isolate the influence of the Indonesian peatland fires from the background changes induced by JAMOC, the changes from the Indonesian fires due to the

in-cloud OVOC oxidation are calculated following

$$\Delta \text{JAMOC} = (\text{FIR}_{\text{JAMOC}} - \text{FIR}) - (\text{REF}_{\text{JAMOC}} - \text{REF}). \quad (1)$$

### 7.1 On the lower troposphere

Figure 15 shows the changes in the zonal mean concentration over Indonesia and the Indian Ocean of all OVOCs explicitly reacting in JAMOC ($\sum$OVOCs; see Eq. B1 in Appendix B) for the simulations without JAMOC (Fig. 15a) and the predicted changes due to JAMOC (Fig. 15b; calculated using Reaction 1), focusing on the Indonesian fire period (SON). Due to the high solubility of many OVOCs and their in-cloud oxidation, their concentration is strongly reduced at altitudes that are characterised by frequent cloud events. Table 5 provides the SEA burden changes for a selection of species that are represented in JAMOC. Overall, the additional in-cloud sink results in a more limited increase in their predicted burden (e.g. only about 87.9 Gg instead of 112.3 Gg for methanol). However, their predicted burden in the reference simulation (REF$_{\text{JAMOC}}$) is also significantly lower, which results in a relative change which is of a similar order as if no in-cloud OVOC oxidation were taken into account.

Figure 16 shows the probability density function (PDF) for EMAC's methanol column bias when compared to IASI satellite retrievals (Franco et al., 2018) in SEA during the Indonesian peatland fires. Without VOC emissions from biomass burning, methanol is slightly underestimated by simulation REF. This underestimation is more pronounced when the in-cloud oxidation of OVOCs is taken into account (simulation REF$_{\text{JAMOC}}$). In both cases, EMAC tends to strongly underestimate methanol in some regions. When VOC biomass burning emissions are taken into account (simulation FIR), these underpredictions are resolved. However, now EMAC tends to strongly overestimate methanol mainly close to biomass burning sources (not shown). These overpredictions are reduced once in-cloud OVOC oxidation is implemented (simulation FIR$_{\text{JAMOC}}$). A high fraction of SEA is

**Table 5.** The SEA tropospheric burden during the Indonesian fire period with and without VOC biomass burning emissions of OVOCs explicitly reacting in JAMOC. The tropospheric burden is given in gigagrams (Gg).

| Species | REF | $\Delta$FIR | Rel. [%] | REF$_{JAMOC}$ | $\Delta$FIR$_{JAMOC}$ | Rel. [%] |
|---|---|---|---|---|---|---|
| Methanol | 185.2 | 112.3 | 60.7 | 133.8 | 87.9 | 65.7 |
| Methyl hydroperoxide | 140.8 | 10.7 | 7.6 | 71.7 | 4.0 | 5.6 |
| Hydroxymethylhydroperoxide | 5.0 | 1.0 | 20.2 | 3.4 | 0.6 | 17.0 |
| Ethanol | 9.1 | 1.0 | 10.5 | 7.1 | 0.8 | 11.5 |
| Ethylene glycol | 0.2 | 0.2 | 73.6 | 0.1 | 0.1 | 70.1 |
| Glycolaldehyde | 25.6 | 17.2 | 67.1 | 13.2 | 10.6 | 80.8 |
| 1-Hydroperoxyacetone | 4.7 | 0.7 | 14.5 | 2.4 | 0.3 | 13.2 |
| Methyl glyoxal | 21.5 | 2.2 | 10.2 | 15.5 | 1.7 | 10.9 |
| Isopropyl hydro peroxide | 0.8 | 0.1 | 18.1 | 0.4 | 0.1 | 17.3 |

covered by oceans. Millet et al. (2008) suggested that some regions of the Pacific and Indian Ocean are a net source of methanol. As discussed by Rosanka et al. (2020b), EMAC represents the ocean as a net methanol sink. Therefore, when comparing the predictions of methanol from EMAC to satellite observations, a certain underestimation is expected. Thus, simulation FIR$_{JAMOC}$ compares the best with IASI retrievals, since it has overall the lowest relative biases.

Changes in hydrocarbons are minimal due to their low solubility, whereas strong changes are predicted for the relatively insoluble $O_3$. Due to in-cloud OVOC oxidation, the initially predicted increase in $O_3$ in western Indonesia and over the Indian Ocean (Sect. 5.4) is dampened by more than 60 % once JAMOC is implemented. This limited increase is caused by the importance of clouds as an $O_3$ sink. This process is globally analysed by Rosanka et al. (2020b) and is based on the enhanced $HO_2$ formation in cloud droplets by OVOC oxidation. Within clouds, $HO_2$ is in acid equilibrium with the superoxide anion ($O_2^-$), which actively destroys $O_3$.

## 7.2   On the lower stratosphere

As seen in Fig. 15b, the in-cloud OVOC oxidation leads to the reduction of their concentrations in the UTLS. Table 6 presents the lower stratospheric burden changes in November due to JAMOC. Overall, the in-cloud oxidation of the OVOCs leads to a more limited increase in their concentration in the lower stratospheric burden induced by the biomass burning emissions. For example, the increase in the methanol burden is limited to 2.6 Gg (instead of 5.8 Gg). In the case of ethylene glycol ($HOCH_2CH_2OH$), the lower stratospheric burden decreases even by about 85 % when JAMOC is used, with and without VOC biomass burning emissions taken into account. Similarly to the changes in the lower stratosphere, the relative change for the simulation using JAMOC is of a similar order as for the simulation without VOC biomass burning emissions but tends to be slightly lower for some OVOCs. This is especially the case for isopropyl hydro peroxide (($CH_3$)$_2$CHOOH), which is lower by about 23 %. The increase in the lower stratospheric phenol concentrations is

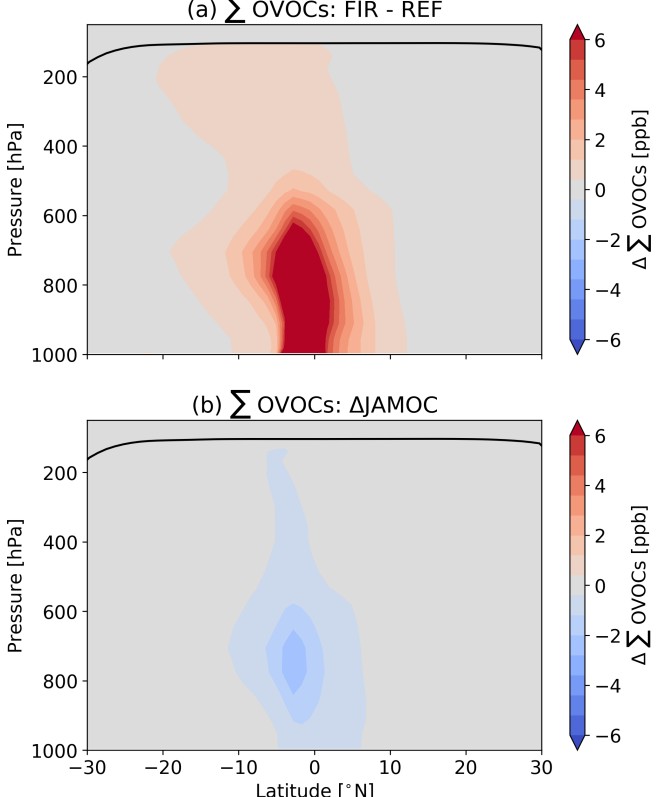

**Figure 15.** Mean zonal change in the sum of all OVOCs explicitly reacting in JAMOC ($\sum$OVOCs; see Eq. B1 in Appendix B) over Indonesia and the Indian Ocean during the 2015 Indonesian fire period (SON). **(a)** Changes due to VOC biomass burning emissions (difference between simulation FIR and REF) and **(b)** changes due to JAMOC ($\Delta$JAMOC).

only slightly impacted and decrease by about 13 % (from 167 % originally; see Table 4) and thus has only little impact on the destruction of $O_3$ by phenoxy radicals. This is consistent with the main source of phenol being the oxidation of benzene, which has a lifetime of the order of 1–2 weeks.

**Table 6.** The lower stratospheric burden in November with and without VOC biomass burning emissions of OVOCs explicitly reacting in JAMOC. The stratospheric burden is given in megagrams (Mg).

| Species | REF | $\Delta$FIR | Rel. [%] | REF$_{JAMOC}$ | $\Delta$FIR$_{JAMOC}$ | Rel. [%] |
|---|---|---|---|---|---|---|
| Methanol | 17 097.87 | 5821.01 | 34.05 | 10 102.29 | 2634.34 | 26.08 |
| Methyl hydroperoxide | 637.88 | 55.45 | 8.69 | 562.11 | 28.3 | 5.03 |
| Hydroxy methyl hydroperoxide | 60.56 | 12.14 | 20.04 | 49.16 | 6.17 | 12.55 |
| Ethanol | 45.35 | 15.5 | 34.18 | 24.15 | 5.45 | 22.55 |
| Ethylene glycol | 0.64 | 0.24 | 38.47 | 0.09 | 0.03 | 31.04 |
| Glycolaldehyde | 84.23 | 16.41 | 19.48 | 54.31 | 11.73 | 21.61 |
| 1-Hydroperoxyacetone | 10.83 | 3.86 | 35.59 | 5.96 | 1.76 | 29.57 |
| Methyl glyoxal | 6.58 | 0.34 | 5.11 | 3.5 | 0.2 | 5.6 |
| Isopropyl hydro peroxide | 2.61 | 1.9 | 72.5 | 2.1 | 1.03 | 49.06 |

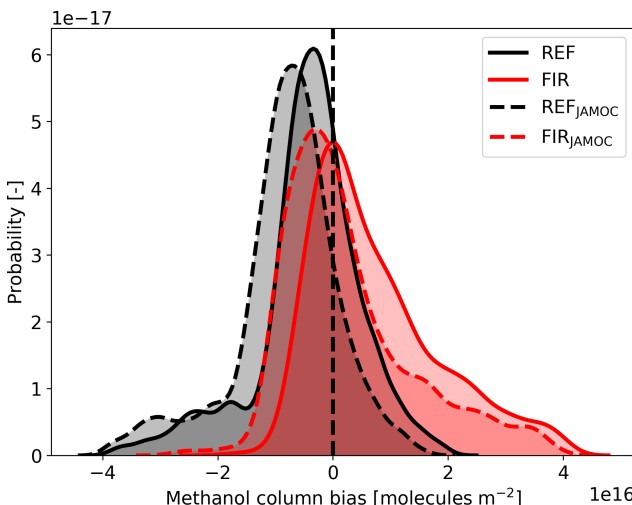

**Figure 16.** Probability density function of EMAC's methanol column bias compared to the IASI satellite measurements for simulation REF, FIR, REF$_{JAMOC}$, and FIR$_{JAMOC}$ in SEA during the 2015 Indonesian peatland fires.

To conclude, in-cloud OVOC oxidation is important to properly represent the resulting impacts from strong pollution events especially during the monsoon season. Overall, the predicted impact on VOCs, radicals, and O$_3$ is dampened by the in-cloud oxidation and models neglecting this process probably tend to overestimate the impact of such an event. It is widely recognised that clouds may act as a source of secondary organic aerosols (SOAs) due to in-cloud oxidation processes (Blando and Turpin, 2000; Ervens et al., 2011; Ervens, 2015). Ervens et al. (2011) suggested that cloud processes might contribute to SOA formation on the same order as gas-phase processes. Within this study, SOA formation from cloud processes is not explicitly represented. However, it is expected that the enhanced VOC concentrations from biomass burning will lead to an increased SOA formation from aqueous-phase processes due to the enhanced formation of oligomers (e.g. from glyoxal and methyl glyoxal) within clouds.

# 8 Model uncertainties

The most important aspects that influence our results are the representation of the transport processes, using different model resolutions, and the chemical kinetics. Each aspect is associated with some uncertainties, which are all shortly discussed in this section.

The magnitude of the depletion in lower stratospheric O$_3$ depends closely on the representation of the vertical transport that conveys the emitted VOCs into the UTLS. In order to evaluate the vertical transport processes of global models, $^{222}$Radon ($^{222}$Rn, radioactive decay half-life of 3.8 d) is typically used (Mahowald et al., 1997; Zhang et al., 2008; Jöckel et al., 2010). Jöckel et al. (2010) and more recently Brinkop and Jöckel (2019) analysed the ability of EMAC to capture the $^{222}$Rn surface concentrations and vertical profiles. Their findings indicate that the vertical transport is well represented in EMAC (using the T42L90MA resolution) and that they are comparable to the earlier analysis with ECHAM5 (the base model of EMAC) by Zhang et al. (2008). Figure 4 shows that the horizontal transport is also an important aspect that influences the distribution of the emitted VOCs from Indonesian peatland fires. Evaluating the horizontal transport using observations (like $^{222}$Rn) is currently not possible. Recently however, Orbe et al. (2018) compared transport timescales of various global models, including EMAC. They found that the horizontal transport from NH mid-latitudes to the tropics differs by 30 %. Based on this comparison, it can be assumed that the horizontal transport is reasonably well represented in EMAC.

In this study, we transfer our process understanding from the fine-resolution (T106L90MA) to the coarse-resolution (T42L90MA) simulations. It is therefore important to understand how well transport processes agree between both resolutions. Currently, no direct analysis has been performed that focuses on the impact of different resolutions on transport processes in EMAC. However, Aghedo et al. (2010) analysed the influence of different horizontal and vertical resolutions in ECHAM5. Since EMAC uses the same horizontal and vertical transport scheme as ECHAM5, we assume that

their findings also apply to EMAC. They find that the vertical transport mainly depends on the number of levels used. By increasing the number of layers from 19 to 31 levels, the mass transported into the stratosphere reduces globally by about 36 %, whereas increasing the resolution from T42 to T106 only decreases the vertically transported mass globally by about 10 %. Here, the influence is the lowest (about 7 %) at high latitudes and the highest in the tropics (about 17 %). Aghedo et al. (2010) suggested that the higher impact in the tropics is probably related to tropical convection processes. Increasing the resolution changes the meridional transport in most regions by less than 2 % and is thus negligible. For our purposes, differences in the inter-hemispheric transport are also negligible. The mean transport time from the NH to SH decreases from 11.9 to 11.8 months and for the SH to NH transport from 11.4 to 11.5 months when increasing the horizontal resolution from T42 to T106. By using the same vertical resolution (90 levels), the highest uncertainty introduced by using different resolutions is eliminated. It is therefore expected that the important transport processes are comparable and properly represented in both resolutions.

We find that the reaction of phenoxy radicals with $O_3$ (Reaction R2) has a significant influence at the surface, in the troposphere, and in the lower stratosphere. As discussed by Taraborrelli et al. (2021), the chemical kinetics used in MOM to represent this $O_3$ loss is associated with some uncertainties. Currently, only the measured reaction rate constant for $C_6H_5O$ is available, and this is used for all phenoxy radicals. Yet, no experimental evidence has been found for the formation of phenyl peroxy radical ($C_6H_5O_2$), which might influence the cycling nature of this $O_3$ loss by Reactions (R1) and (R2). However, this product is still to be expected. Even with different products, a significant depletion of $O_3$ is anticipated by Reaction (R2). At the same time, the reaction rate from Tao and Li (1999) is reported to be at the lower end, whereas a higher reaction rate would increase the depleted $O_3$. Additionally, Taraborrelli et al. (2021) report that MOM neglects the non-HONO formation channel from nitrophenol photolysis, which does not destroy the aromatic ring and re-forms phenoxy radicals (Cheng et al., 2009; Vereecken et al., 2016). It is therefore expected that, due to increasing nitrophenol concentrations in the lower troposphere (Sect. 5.3) as well as in the UTLS, the importance of Reaction (R2) as an $O_3$ sink is potentially underestimated.

the emissions from the Indonesian peatland fires and global biomass burning events.

Our results indicate that VOC emissions from biomass burning are important to reproduce hydrocarbons and secondary OVOCs in the atmosphere. Compared to other biomass burning regions, a particularly strong increase is modelled in the SEA region, due to the unique emission footprint from the Indonesian peatland fires. The enhanced formation of nitrophenols and strong HNCO emissions create toxic conditions in most parts of Indonesia, directly influencing its population. Regionally, significant changes in radical concentrations ($HO_x$ and $NO_x$) are predicted. In general, $O_3$ increases in the lower troposphere with the highest changes in the NH high latitudes due to strong fires in boreal Asia. However, on a global scale, tropospheric changes in $O_3$ are negligible. High aromatic emissions from peatland fires lead to a depletion of $O_3$ in eastern Indonesia.

The ongoing ASMA and the general tropical upward transport during the Indonesian fires lift the emitted VOCs and their oxidation products quickly to the lower stratosphere. Here, especially large increases are predicted for levels of the aromatic compounds benzene and toluene. The oxidation of VOCs results in the reduction of OH and $NO_x$ and the increase in $HO_2$. Additionally, the Indonesian fires becomes a major source of lower stratospheric $NO_3$. Indonesian fires enhance the $O_3$ destruction by phenoxy radicals by up to 20 % in the southern tropical UTLS. This chemical loss propagates into the lower stratosphere and potentially influences the variability of $O_3$ retrieved from satellite observations. Overall, the highest changes in lower stratospheric radicals during the period between 2001 to 2016 are predicted for particularly strong El Niño years, due to strong Indonesian peatland fires.

The overall impact of Indonesian fires on the composition of the troposphere and lower stratosphere is reduced when in-cloud OVOC oxidation is taken into account. In particular, the predicted $O_3$ increase in the troposphere is dampened due to enhanced destruction of $O_3$ within clouds. This suggests that models neglecting the in-cloud oxidation of OVOCs probably tend to overestimate the impact of such an event like the Indonesian peatland fires.

## 9   Conclusions

In this study, the influence of VOC emissions from reoccurring Indonesian peatland fires is analysed with the main focus on 2015, a particularly strong year. This is achieved by performing multiple global simulations using EMAC. By comparing EMAC's prediction of HCN and CO columns to IASI satellite retrievals, we show that EMAC properly represents

## Appendix A: HCN retrievals from IASI observations

The spaceborne data of HCN columns used in this study are obtained from the IASI radiance spectra by applying the version 3 of the Artificial Neural Network for IASI (ANNI) retrieval framework. Initially developed for the retrieval of $NH_3$ and dust from the IASI observations (Whitburn et al., 2016a; Clarisse et al., 2019), ANNI v3 incorporates updates and modifications to allow the retrieval of a suite of VOCs. Until now, it has been used to retrieve methanol, formic acid, and PAN (Franco et al., 2018) and then acetone (Franco et al., 2019) and acetic acid (Franco et al., 2020). Here, we perform the HCN retrieval by applying the full ANNI v3 procedure. As this approach has already been described in detail (see Franco et al., 2018, and references therein), we limit ourselves here to a summary of the main retrieval steps and to the elements specific to the retrieval of HCN. Examples of HCN columns from IASI single overpasses in the 2015 Indonesian fire plumes and averaged distributions are also presented.

As mentioned in Sect. 2.1.3, the ANNI retrieval method proceeds in two major steps. First, in each individual IASI radiance spectrum, the target species is detected and the strength of its absorption is quantified by a metric called the hyperspectral range index (HRI). Then, the HRI is converted into a gas total column by means of an artificial feedforward neural network (NN), which also provides an uncertainty on the retrieved column.

The HRI is a dimensionless metric of the magnitude of the spectral signature of a target species in a given IASI spectrum, relative to the spectral variability of a "background" atmosphere in the absence of the target gas, i.e. a variability resulting from all other parameters that contribute to the spectral radiance, such as other atmospheric gases (see Walker et al., 2011). The HRI is calculated over the main spectral range, in which the target species absorbs. The HCN absorption band ($\nu_2$ branch) included in the IASI spectrum is situated close to a strong $Q$ branch of $CO_2$ near $720\,cm^{-1}$. Therefore, the whole $700$–$800\,cm^{-1}$ spectral range covering many HCN features is used to calculate the HRI. The $CO_2$ line mixing in that range is accounted for as described by Duflot et al. (2013). A first HRI of HCN was already set up for the IASI observations by Duflot et al. (2015), but here we set up a new more sensitive one following the iterative procedure presented by Franco et al. (2018).

In contrast to Duflot et al. (2015), who used pre-calculated coefficients to link the HRI to the HCN total column, the ANNI v3 procedure implements an artificial feedforward NN for this purpose. Such a NN is set up to mimic in a comprehensive way the complex connections that exist between the HRI, the state of the atmosphere and Earth's surface, and the gas abundance. Setting up a NN requires a training phase in which the NN learns from the presentation of an extensive dataset including all the necessary input and output variables. In ANNI v3, the NN inputs are the HRI, a spectral baseline temperature, the $H_2O$ columns, the temperature pro-

file, the surface pressure and emissivity, and the IASI viewing angle, whereas the output is the HCN column. Here, we built this training set from over $250\,000$ synthetic IASI spectra simulated by a line-by-line radiative transfer model. The advantage of such a synthetic training set is that it is free of the noise and/or scarcity of real measurements and that the spectra can be generated in large amounts in order to make the training set – and hence the NN – representative of all possible conditions. For example, the NN set up for HCN is trained to retrieve gas column from $1 \times 10^{14}$ to $15 \times 10^{16}$ molecules $cm^{-2}$. Actually, two separate synthetic datasets are assembled per target species, one being representative of conditions close to emission sources and the other of mixing/transport conditions (see Whitburn et al., 2016a; Franco et al., 2018, for the rationale). Each training set leads to the setup of a specific NN that is used to globally retrieve the target species in emission or transport regimes, successively. The training performances are similar to those of the other VOCs retrieved with ANNI v3 and are reached with a NN made of two computational layers, with each layer deploying eight nodes.

In addition to the total column, the NN returns an associated error that is calculated via a perturbation method of the input variables (see Whitburn et al., 2016a). A pre-filter prevents the retrieval on cloudy scenes (cloud coverage $> 10\,\%$) or for observations with missing ancillary data. Consistent with the other ANNI VOCs products, a post-filter discards the individual retrievals affected by uncertainties that are too large or by poor measurement sensitivity to HCN, specifically when

$$\left| \mathrm{column_{(HCN)}} \Big/ \mathrm{HRI_{(HCN)}} \right| > 8 \times 10^{15}\ \mathrm{molecules\,cm^{-2}} \quad \text{(A1)}$$

or spectral baseline temperatures $< 268\,K$. This post-filter is not (directly) driven by the gas abundance but rather by the thermal contrast (Franco et al., 2020). Finally, the constant climatological background of target gas abundance that is not accounted for by the HRI has been estimated as $1.85 \times 10^{15}$ molecules $cm^{-2}$ for HCN (see Franco et al., 2018); this offset is thus added to the individual retrieved columns. Once set up, the NN is fed for each individual IASI observation with the appropriate input data. Here, we chose to use the ERA-5 reanalysis dataset (Hersbach et al., 2020) for the meteorological input data in the network. In the framework of the evaluation of EMAC in the 2015 Indonesian fires (see Sect. 4), only the HCN product obtained with the NN in transport/mixing regime has been exploited. Indeed, the Cloud-Aerosol Lidar with Orthogonal Polarization (CALIOP) aboard CALIPSO indicates the fire plume located in the free troposphere during this massive biomass burning event.

Figure A1 presents the daily distributions of HCN total columns from IASI/Metop-A and B observations in South East Asia, for 6 successive days taken during the 2015 Indonesian fires. Whereas background ar-

eas are characterised by HCN total columns generally lower than $0.5 \times 10^{16}$ molecules cm$^{-2}$, on the first day (29 September 2015), strong HCN enhancements ($> 4 \times 10^{16}$ molecules cm$^{-2}$) are detected by IASI in the vicinity of Sumatra, indicating a massive fire plume. After 6 d (4 October 2015), we can observe that the plume has grown progressively and that the bulk of HCN has been transported to the west across the Indian Ocean. The retrieved column uncertainties in the area generally fall in the range of $2$–$5 \times 10^{15}$ molecules cm$^{-2}$. Note that these uncertainties are reduced significantly by averaging numerous IASI measurements to build monthly or seasonal mean distributions of HCN columns. The typical seasonal distributions of IASI/Metop-A HCN columns are presented in Fig. A2 for the 2011–2014 time period, i.e. for years without massive fire events, such as the 2010 Russian fires or the 2015 Indonesian fires. These distributions highlight the dominant contribution of biomass burning to the atmospheric HCN burden, with HCN enhancements detected in Africa throughout the year; in South East Asia in March–April–May; in India, eastern China, and the Northern Hemisphere mid- and high latitudes during the boreal summer; and within the tropics in September–October–November. Important outflows from these source regions are also noticeable, especially over the oceans. Figure A3 presents the monthly mean HCN columns during the 2015 Indonesian fires (from September–December) along with the corresponding distributions over the 2011–2014 time period. It illustrates the exceptional intensity of the 2015 fires compared to the previous years, with important HCN enhancements detected throughout the entire intertropical band.

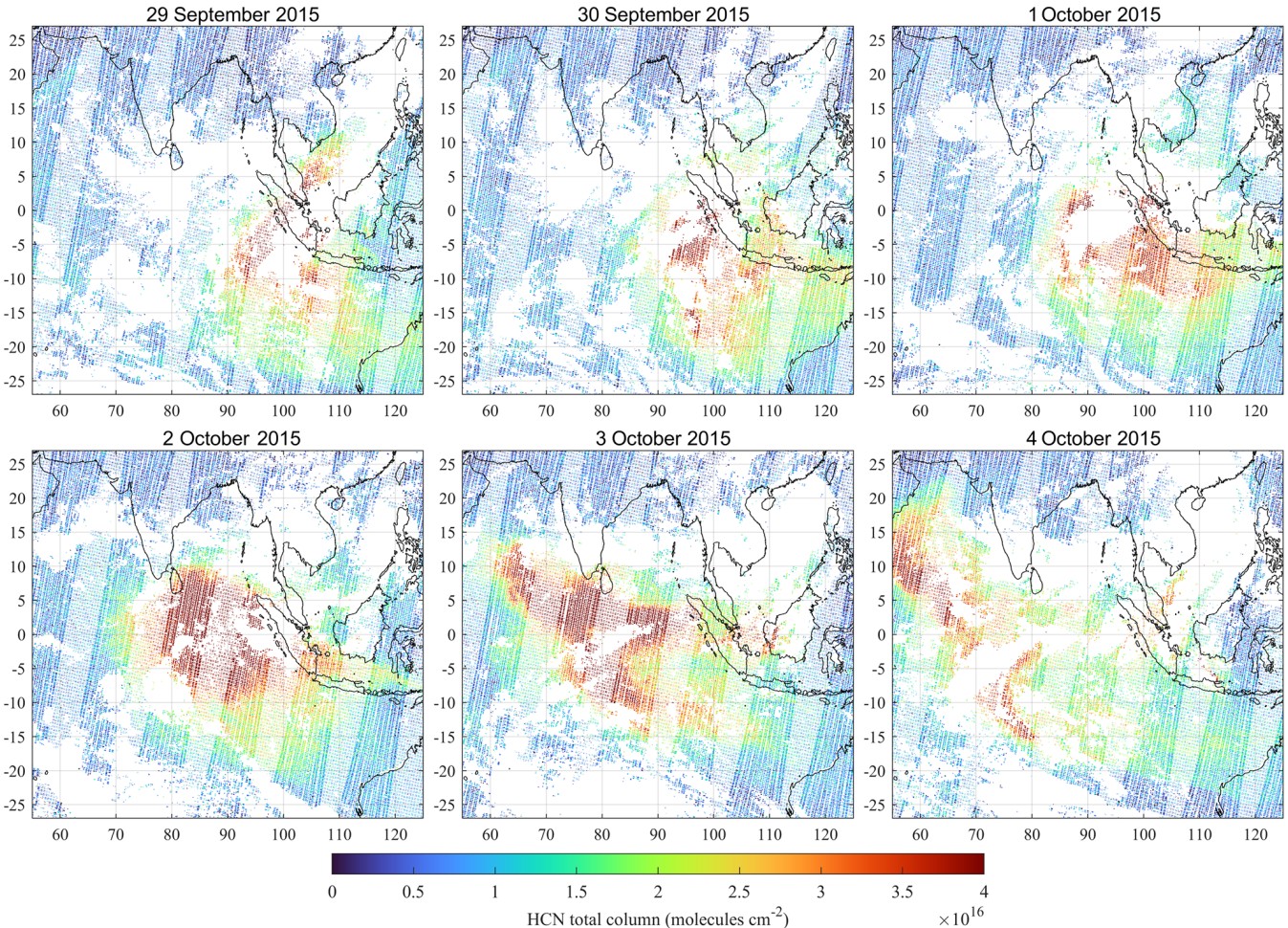

**Figure A1.** Daily regional distributions of HCN total column (in molecules cm$^{-2}$) derived from the IASI spectra recorded in the morning overpasses of Metop-A and B, for 6 successive days during the 2015 Indonesian fires. These distributions take into account the actual footprint on the Earth's surface of each individual IASI measurement, i.e. a small circle at nadir and an elongated ellipse at the limit of the across-track swath of the satellite. Note the complementarity of the IASI/Metop-A and B flight tracks that avoid gaps between the successive overpasses in the tropics. The white areas correspond to data filtered out because of unsatisfactory retrieval quality or the presence of clouds.

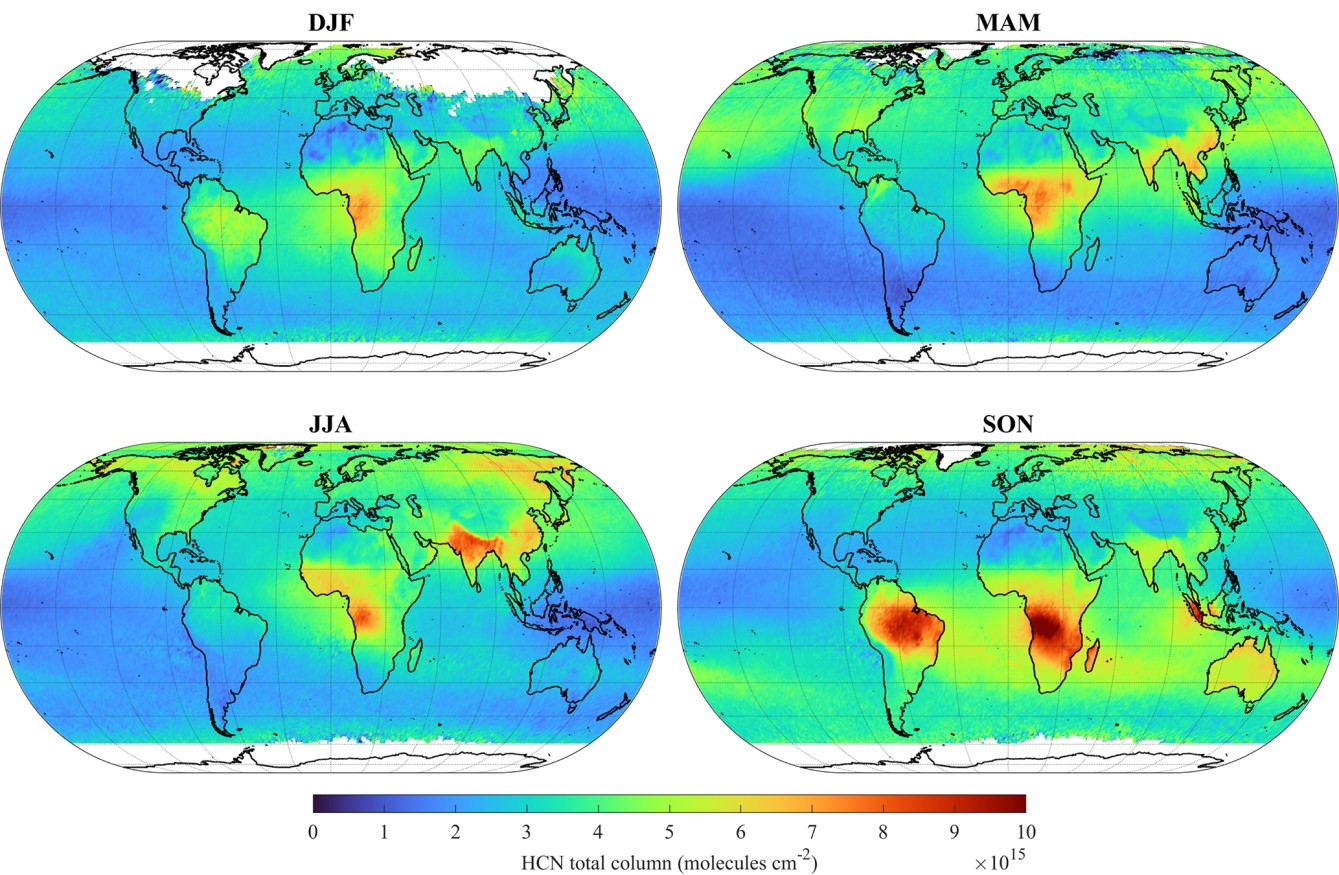

**Figure A2.** Seasonal means (on a $1 \times 1°$ grid) of the HCN total columns (in molecules cm$^{-2}$) retrieved from the IASI/Metop-A measurements over the 2011–2014 time period. The HCN columns over the continents have been retrieved with the NN in emission regime, whereas the NN in transport/mixing regime has been used over the oceans.

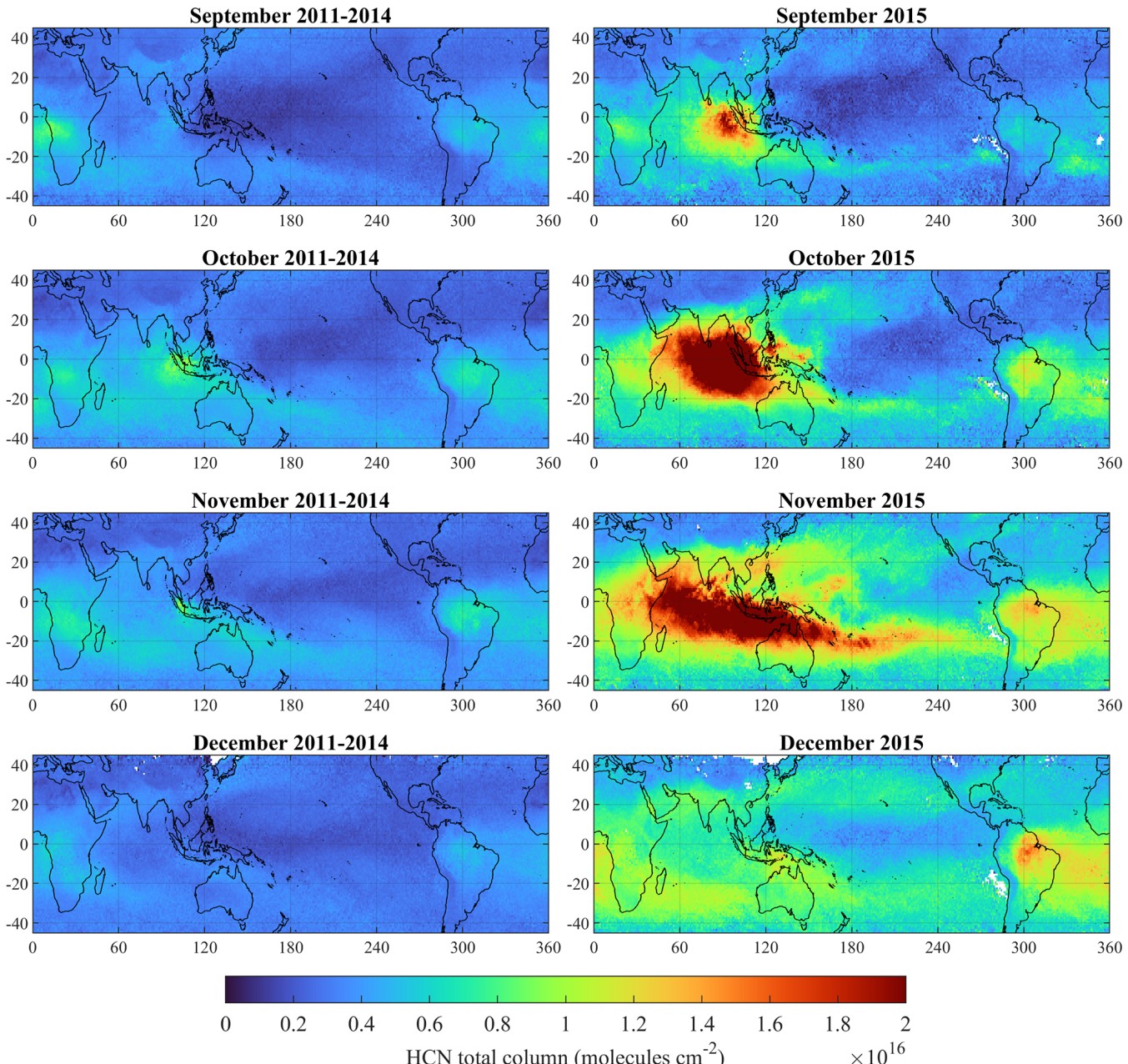

**Figure A3.** Monthly means (on a $1 \times 1°$ grid) of the HCN total columns (in molecules $cm^{-2}$) retrieved from the IASI/Metop-A measurements over the 2011–2014 time period (left plots) and over the year 2015 (right plots).

**Appendix B:  Definition of $\sum$OVOCs**

In Fig. 15, the mixing ratios of the sum of all the OVOCs explicitly reacting in JAMOC ($\sum$OVOCs) are shown. In this case, $\sum$ OVOCs is defined as

$$
\begin{aligned}
\sum \text{OVOCs} = {} & \text{methanol} + \text{formaldehyde} \\
& + \text{methyl hydroperoxide} \\
& + \text{hydroxymethylhydroperoxide} \\
& + \text{ethanol} + \text{ethylene glycol} \\
& + \text{acetaldehyde} + \text{glycolaldehyde} \\
& + \text{glyoxal} + \text{1-hydroperoxyacetone} \\
& + \text{methylglyoxal} + \text{isopropanol} \\
& + \text{isopropyl hydro peroxide} \\
& + \text{methacrolein} + \text{methyl vinyl ketone}. \quad \text{(B1)}
\end{aligned}
$$

*Data availability.* The simulation results are archived at the Jülich Supercomputing Centre (JSC) and are available upon request. The IASI VOC columns retrieved with the ANNI framework are available upon request. The IASI CO data processed with FORLI-CO v0151001 can be downloaded from the AERIS portal at http://iasi.aeris-data.fr/CO/ TS3 (last access: 22 February 2021).

*Author contributions.* The study was designed by SR and DT. AP implemented the algorithms for VOC emissions from biomass burning and terrestrial vegetation. SR adjusted the biomass burning emission factors. AP and DT implemented HCN deposition to the ocean. SR performed the simulations and analysed the data with contributions from DT. BF, LC, and PFC developed the IASI VOC products and contributed to the analyses. SR and DT discussed the results with contributions from BF and AW. The manuscript was prepared by SR with the help of all co-authors.

*Competing interests.* The authors declare that they have no conflict of interest.

*Special issue statement.* This article is part of the special issue "The Modular Earth Submodel System (MESSy) (ACP/GMD interjournal SI)". It is not associated with a conference. TS4

*Acknowledgements.* The work described in this paper has received funding from the Initiative and Networking Fund of the Helmholtz Association through the project Advanced Earth System Modelling Capacity (ESM). The content of this paper is the sole responsibility of the author(s), and it does not represent the opinion of the Helmholtz Association, and the Helmholtz Association is not responsible for any use that might be made of the information contained. The authors gratefully acknowledge the Earth System Modelling project (ESM) for funding this work by providing computing time on the ESM partition of the supercomputer JUWELS at the Jülich Supercomputing Centre (JSC). IASI is a joint mission of EUMETSAT and the Centre National d'Etudes Spatiales (CNES, France). The authors acknowledge the AERIS data infrastructure for providing access to the IASI data, Daniel Hurtmans for the development of the CO retrievals, and EUMETSAT AC SAF for CO data production. The research at ULB has been supported by the project OCTAVE (Oxygenated Compounds in the Tropical Atmosphere: Variability and Exchanges, http://octave.aeronomie.be/, last access: TS5) of the Belgian Research Action through Interdisciplinary Networks (BRAIN-be; 2017–2021; research project BR/175/A2/OCTAVE) and by the IASI.Flow Prodex arrangement (ESA–BELSPO). Lieven Clarisse is a research associate supported by the F.R.S.–FNRS.

*Financial support.* TS6 This research has been supported by the Initiative and Networking Fund of the Helmholtz Association through the project Advanced Earth System Modelling Capacity (ESM) (grant no. DB001549).

The article processing charges for this open-access publication were covered by the Forschungszentrum Jülich.

*Review statement.* This paper was edited by Bryan N. Duncan and reviewed by two anonymous referees.

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

TS3    Please provide a direct link to the data set and, if possible, a DOI instead of a URL. In any case, please provide a reference list entry including creators, title, and date of last access.

TS4    Please confirm.

TS5    Please provide date of last access (day month year).

TS6    Please note that there is a discrepancy between funding information provided by you in the acknowledgements and the funding information you indicated during manuscript registration, which we used to create this section. Please double-check your acknowledgements to see whether repeated information can be removed from the acknowledgements or changed accordingly. If further funders should be added to this section, please provide the funder names and the grant numbers. Thanks.

TS7    Please provide pages or article number.

TS8    Please provide pages or article number.

TS9    Please provide volume and pages or article number.

TS10    Please provide pages or article number.

TS11    Please provide pages or article number.

TS12    Please provide date of last access (day month year).

TS13    Should this dataset be added to the data availability section?

TS14    Please provide pages or article number.