# Peer review of "The impact of organic pollutants from Indonesian peatland fires on the tropospheric and lower stratospheric composition"

_Atmospheric Chemistry and Physics, 2020_

## Referee Comment (RC1) · Anonymous Referee #1 · 31 Dec 2020

Rosanka et al., 2020 presented a modeling study to quantify the impact of Indonesia peatland fire burning on atmospheric chemical composition. Overall, I found this an interesting study with well-designed model simulations. Use model sensitivity simulations to address the impact of aqueous phase chemistry of fire emissions is a somewhat new area that haven't been discussed much in literature. However, the paper in its present form needs some major improvements, including better figures, more in-depth analysis and uncertainty discussion. These concerns need to be addressed before the paper be accepted for publication in ACP.

1. Title of this paper, in my view, is not accurate in describing the content of what's

presented in this work: i) this study primarily targets the Indonesia peatland fires, the use of word "tropical" implies the entire equatorial tropical band, which is misleading to some extent even though I understand a large fraction of the tropical peatland fires are from SE Asia, ii) most of the results from this work focused on the impact of peatland fires on tropospheric VOC, NOx, OH, O3; the impact on lower stratospheric ozone is just a minor component of this work and I am not convinced the impact is important nor statistically significant compared to the potential dynamical variability. Hence the use of "... its impact on lower stratospheric ozone" is not appropriate.

2. The paper is in general well written. However, I did find many grammatic and editorial mistakes during reading. I tried to include most of these down below, but I am sure there are more places need correction. I would strongly encourage the authors have a full editorial read of the manuscript before resubmission.

3. L68-L90. I think this is a very helpful paragraph and a useful discussion in putting the Indonesian biomass burning in the context of other biomass burning emissions regions around the globe. But I am not sure it belongs to an "Introduction" section. Is it possible to move it to the main results section? If you are strongly inclined to keep it in the introduction, this section will probably flow much better if you move this paragraph before para 2 (Asian Monsoon).

4. Figure 4 and the related discussion of model bias in simulated HCN in Section 3. The minimum and maximum values used for the color scales not adequate for readers to assess model performance. For example, IASI column saturates at $2\times10^{16}$ while the model FIR run high bias over the same region also saturates at $+2\times10^{16}$, and the model REF run have negative biases that also saturate at $-2\times10^{16}$. You need to increase the saturation values before the readers can reasonably assess the model performance and understand the magnitudes of emissions biases we are looking at over Indonesia. Without a properly done Figure 4, I find the discussion in section 3 hand waving and not convincing. With that said, with the difference between REF and FIR roughly two times the observed HCN amount over Indonesia, I strongly suspect

the emissions and/or emission factors used for biomass burning being highly biased. Although CO is not a unique tracer to biomass burning, since (a) remote sensing measurements of CO is available from multiple sensors and retrievals are available with much higher precision, (b) SE Asia during biomass burning season is most likely dominated by CO from biomass burning, I would strongly encourage the authors to check the simulated CO from these two simulations and how do they compare with satellite measurements, e.g. from IASI, to see if such high bias issues existed for CO as well.

5. I think Section 4 has a lot of interesting results regarding the impact of the 2015 Indonesian peatland fires. I strongly suggest the authors consider summarize these results into a schematic diagram that illustrates (1) the direct impact on primarily emitted gases, e.g. C2H2, C2H6, (2) changes in species such as phenol and methanol, due to direct and secondary production from primary VOC oxidation, (3) the subsequent impact on OH, NOx, NO3, O3. You have the quantitative numbers calculated and given in the text already. It will be much more useful for readers if these information are assembled into an easy-to-understand diagram and will definitely improve the paper.

6. My biggest concern with this analysis is I am not convinced that the impact of Indonesian peatland fires on lower stratospheric O3 are as significant as the authors claimed. More thorough analysis is needed before one should jump to such conclusions. Here are a few of my reasons for saying so:

i) The FIR-REF difference (ïĄĎO3) for simulated O3 at 50 hPa in the tropics is on the order of 10 ppb (max = - 12 ppb). Compared to typical O3 concentrations of 1-2 ppm in this region, this is a change of about 0.5-1% even in the maximum change region. Consider the high uncertainty and likely high biases in VOC emissions from peatland burning (as pointed out in my comment #4) and large uncertainties in aqueous phase chemistry, even this small impact could be a high-biased impact.

ii) From Figure 14 and the related discussion in Section 6, I am not sure how did

you calculated and arrived at the -0.38 DU change in lower stratospheric O3. There are large dynamically-driven and chemically-driven variabilities of tropical lower stratospheric O3, at minimum you need to performance a statistically robust trend analysis, describe what you did, show your trend analysis results and discuss the results. In addition to just showing the differences, I would strongly encourage you compare the simulated total lower stratospheric O3 from these two simulations, analyze this change in the context of dynamical-driven variabilities in O3 before concluding such change is significant and the related aqueous process need to be considered in chemical modeling.

iii) By looking at Figures 8, 13, 14, with the documented ozone abundance and variability in previous literature, the change in tropospheric and lower stratospheric O3 are too small, in my opinion, to suggest VOC emissions from Indonesian fires can change atmospheric O3 in a significant way, except in the small localized regions near Indonesia.

Minor comments:

L12. hydroxyl radicals (OH) → the hydroxyl radical

L13-14. Suggested rewording: While an increase in ozone is predicted close to the peatland fires, simulated O3 decreases in eastern Indonesia due to particularly high phenols.

L16. "the impact of such extreme pollution events." The impact on what? Please clarify.

L18. Not really appropriate to use "high" destruction. May be substitute it with "large" or "efficient"?

L20-22. What's the statistical significance of such decrease compared to the variability of ozone in this region? See above major comment.

L26. "," missing before "resulting"

L30. Please specify which months are the dry season.

L33. → Gaveau et al. (2014) estimated

L39. Delete "this" before "smouldering fires"

L41. → A major fraction ... is (instead of "are")

L41. → which comprise a large variety of species and can influence atmospheric ...

L48. "This almost stationary globally prevailing meteorological pattern" – too many adjectives! And it is not global either. How about "This semi-stationary large-scale meteorological pattern"?

L58. What do you mean by "usual systems"? Consider use "other meteorological systems"

L76. → where the most of peatland is burned.

L76. How about change "result in" to "have"?

L77. "Indonesia is characterized by a unique emission footprint". It is helpful to readers if you can elaborate the uniqueness of the Indonesian fire emissions, instead of implying it is unique. What's unique about it? You may0 consider move the sentence from L83-84 to here. L103. Suggest use "simulated results with", instead of "prediction to ..."

L109. → provided

L119. Within this ... → For this ...

L119. Suggest change "modelled" to "computed" to avoid redundancy with submodels

L125. You may consider specify "linear hydrocarbons" for readers that are not familiar with this term

L127. I would suggest use "primarily emitted" instead of "heavily emitted"

L150. It is not 100% accurate to say biogenic and biomass burning emissions are the dominant sources of VOCs, consider the importance of anthropogenic emissions. How about change to "two major sources . . ."

L152. "MESSy submodel Model of . . ." reads awkward and redundant. How about use "The MESSy submodel uses Model of . . ."

L243. Delete "even"

L399. Change "O3 sinks" to "an O3 sink"

---

## Referee Comment (RC2) · Anonymous Referee #2 · 9 Jan 2021

This paper presents modeling studies of several aspects of the strong Indonesian biomass burning in 2015, including the impact of VOC emissions, the in-cloud oxidation of VOCs, and the transport of VOCs to the lower stratosphere by the Asian monsoon. This work uses the ECHAM/MESSy model for a series of sensitivity tests, including two horizontal resolutions (1 deg and 2.8 degrees), and a comparison of aqueous-phase mechanisms. A new satellite product, HCN from IASI, is described in the appendix and used for model evaluation. A model experiment where all VOC biomass burning emissions are excluded is the basis for much of the analysis in the paper. There are a number of interesting results and new results, such as the aqueous chemistry impacts, presented, but I agree with the other reviewer that a major revision of the paper

is required before publication.

The paper includes a broad set of topics - tropospheric composition, health impacts, aqueous chemistry, UTLS impacts - which make the paper seem a bit disjointed. As the other reviewer pointed out, the impact on stratospheric ozone appears to be insignificant and I recommend leaving that out.

The paper discusses VOCs quite generally in many places, but does briefly summarize in Section 4.1 various categories of VOCs, and describes their differences. A bit more focus is placed on aromatics, and particularly phenols, which are certainly significant in biomass burning, but a number of other HCs and OVOCs are also important and should be discussed.

The discussion of HCN does not really fit with the reactive chemistry analysis and I do not find the use of it to evaluate the biomass burning emissions in the model very convincing. This paper seems to be an opportunity to describe the new IASI retrievals of HCN, but it seems like that discussion might be appropriate for a short AMT paper, with comparisons to other satellite retrievals of HCN and validation with in situ aircraft observations. Evaluation of the model with the established and validated CO satellite retrievals, from IASI and other platforms, would provide greater confidence in the model simulations.

Here are some of the technical corrections that are needed:

l.110: in the resolution label, L90 apparently indicates 90 levels; explain 'MA'.

l.123: 'capable to represent' should be 'capable of representing'

l.193: 'one simulation exists, in which all ...' -> ' in one simulation all ...'

l. 339: 'destruct' -> 'destroy'

l.348: "80's" -> "1980s"

l.396: relative -> relatively

l.405: rewrite "which even enhances by in-cloud ..."

l.407: 'in the same order to SOA formation" -> "to SOA formation on the same order"

l.431: reference for La Nina strengthening AMSA?

l.431: 'strengthens' -> 'has strengthened'

l.433: extend -> extent

l.456: 'appoint' is not the right word here. suggest? hypothesize?

l.467: 'our study' - does this refer to this manuscript, or previous work? If this paper, where is this shown?

l.506: deletion -> depletion

Table 1: column 2 would be better labeled 'Dominant fuel type' (fire type implies to me flaming or smoldering, for example).

Table 2: What does 'ScSta' mean? Also, define JAMOC here.

[Figure]

---

## Author Comment (AC1) · 12 Apr 2021

**Reply to comments of Anonymous Referee #1**

Rosanka et al., 2020 presented a modeling study to quantify the impact of Indonesia peatland fire burning on atmospheric chemical composition. Overall, I found this an interesting study with well-designed model simulations. Use model sensitivity simulations to address the impact of aqueous phase chemistry of fire emissions is a somewhat new area that haven't been discussed much in literature. However, the paper in its present form needs some major improvements, including better figures, more in-depth analysis and uncertainty discussion. These concerns need to be addressed before the paper be accepted for publication in ACP.

Thank you very much for the helpful comments and seeing the value of our work to the community. Following the comments of both referees, we revised the manuscript significantly focusing on the following points:

- The title of the manuscript was adjusted to better represent the structural changes performed during the revision.

- We moved the comparison of the emission footprint of the Indonesian peatland fires to other biomass burning regions from the introduction to a new dedicated section (now Sect. 3).

- In addition to updating the section on the comparison of EMAC's representation of HCN to IASI satellite retrievals, we now also include a comparison to IASI CO retrievals.

- We restructured the reporting of our findings. It now follows a more systematic manner, in which we first report the impact on the troposphere (now Sect. 5), followed by the impact on the lower stratosphere (now Sect. 6). In both sections, we now focus on hydrocarbons, oxygenated organics, nitrogen containing compounds, and radicals in separate subsections. Afterwards the impact of in-cloud OVOC oxidation on the changes in the troposphere and stratosphere is discussed in a separate section (now Sect. 7).

- We extended the tropospheric analysis to include further hydrocarbons and OVOCs.

- Following the concerns of both referees, we removed the trend analysis on the influence of the Indonesian peatland fires on lower stratospheric $O_3$. Instead, we broadened the discussion on the lower stratosphere (hydrocarbons, oxygenated organics, nitrogen containing compounds, and radicals). In addition, we now focus on the change in the importance of the $O_3$ loss by phenoxy radicals in the UTLS and its potential to contribute to the lower stratospheric $O_3$ variability.

Please find in black the original comments and in red our replies.

1. Title of this paper, in my view, is not accurate in describing the content of what's presented in this work: i) this study primarily targets the Indonesia peatland fires, the use of word "tropical" implies the entire equatorial tropical band, which is misleading to some extent even though I understand a large fraction of the tropical peatland fires are from SE Asia, ii) most of the results from this work focused on the impact of peatland fires on tropospheric VOC, $NO_x$, OH, $O_3$; the impact on lower stratospheric ozone is just a minor component of this work and I am not convinced the impact is important nor statistically significant compared to the potential dynamical variability. Hence the use of "... its impact on lower stratospheric ozone" is not appropriate.

    Following the major structural changes of the revised manuscript and the comments from you and the other referee, we agree that the title of this manuscript needs to be updated as well. We therefore suggest the following new title: "The impact of organic pollutants from Indonesian peatland fires on the tropospheric and lower stratospheric composition"

2. The paper is in general well written. However, I did find many grammatic and editorial mistakes during reading. I tried to include most of these down below, but I am sure there are more

places need correction. I would strongly encourage the authors have a full editorial read of the manuscript before resubmission.

Thank you for listing all these grammatical and editorial mistakes. We performed a full editorial revision before resubmission of the manuscript.

3. L68-L90. I think this is a very helpful paragraph and a useful discussion in putting the Indonesian biomass burning in the context of other biomass burning emissions regions around the globe. But I am not sure it belongs to an "Introduction" section. Is it possible to move it to the main results section? If you are strongly inclined to keep it in the introduction, this section will probably flow much better if you move this paragraph before para 2 (Asian Monsoon).

We agree that this discussion deserves its own section. Therefore, we moved this part from the introduction to its own section (Section 3 in the revised manuscript). In the revised version, we extended the discussion to include an analysis of the characteristics of each biomass burning region during El Niño years, non-El Niño years, and in 2015.

4. Figure 4 and the related discussion of model bias in simulated HCN in Section 3.The minimum and maximum values used for the color scales not adequate for readers to assess model performance. For example, IASI column saturates at 2x10ˆ16 while the model FIR run high bias over the same region also saturates at +2x10ˆ16, and the model REF run have negative biases that also saturate at -2x10ˆ16. You need to increase the saturation values before the readers can reasonably assess the model performance and understand the magnitudes of emissions biases we are looking at over Indonesia. Without a properly done Figure 4, I find the discussion in section 3 hand waving and not convincing. With that said, with the difference between REF and FIR roughly two times the observed HCN amount over Indonesia, I strongly suspect the emissions and/or emission factors used for biomass burning being highly biased. Although CO is not a unique tracer to biomass burning, since (a) remote sensing measurements of CO is available from multiple sensors and retrievals are available with much higher precision, (b) SE Asia during biomass burning season is most likely dominated by CO from biomass burning, I would strongly encourage the authors to check the simulated CO from these two simulations and how do they compare with satellite measurements, e.g. from IASI, to see if such high bias issues existed for CO as well.

Following your comment and the comment of the second referee, we considerably revised the evaluation of EMAC's capabilities to represent the Indonesian peatland fires. This includes a revision of the HCN analysis and an updated Figure 4, in which the saturation for the IASI retrievals was increased to $3 \times 10^{16}$ molecules cm$^{-2}$ and for the comparison plots set to $1.5 \times 10^{16}$ molecules cm$^{-2}$. In addition, we now include a comparison of EMAC's CO to IASI retrievals. Here, we find that EMAC tends to slightly underestimate CO over Indonesia and overestimates CO in South America. We attribute the underestimation in Indonesia to a too low emission coefficient used by EMAC. The overestimation in South America is related to an overestimation of biogenic emissions in this region. Following both analyses, we think that EMAC represents the Indonesian peatland fires in a reasonable manner, especially when considering the exceptional strength of the 2015 Indonesian peatland fires.

5. I think Section 4 has a lot of interesting results regarding the impact of the 2015 Indonesian peatland fires. I strongly suggest the authors consider summarize these results into a schematic diagram that illustrates (1) the direct impact on primarily emit-ted gases, e.g. C2H2, C2H6, (2) changes in species such as phenol and methanol,due to direct and secondary production from primary VOC oxidation, (3) the subsequent impact on OH, NO$_x$, NO$_3$, O$_3$. You have the quantitative numbers calculated and given in the text already. It will be much more useful for readers if these information are assembled into an easy-to-understand diagram and will definitely improve the paper.

We are very grateful for this useful comment! Indeed an illustration summarising the main findings of our work is very useful for the reader. We therefore added such an illustration to the revised manuscript (now Fig. 7).

6. My biggest concern with this analysis is I am not convinced that the impact of Indonesian peatland fires on lower stratospheric O$_3$ are as significant as the authors claimed. More thorough

analysis is needed before one should jump to such conclusions. Here are a few of my reasons for saying so:

i) The FIR-REF difference ($\Delta O_3$) for simulated $O_3$ at 50 hPa in the tropics is on the order of 10 ppb (max = - 12 ppb). Compared to typical $O_3$ concentrations of 1-2 ppm in this region, this is a change of about 0.5-1 % even in the maximum change region. Consider the high uncertainty and likely high biases in VOC emissions from peatland burning (as pointed out in my comment #4) and large uncertainties in aqueous phase chemistry, even this small impact could be a high-biased impact.

ii) From Figure 14 and the related discussion in Section 6, I am not sure how did you calculated and arrived at the -0.38 DU change in lower stratospheric $O_3$. There are large dynamically-driven and chemically-driven variabilities of tropical lower stratospheric $O_3$, at minimum you need to performance a statistically robust trend analysis, describe what you did, show your trend analysis results and discuss the results. In addition to just showing the differences, I would strongly encourage you compare the simulated total lower stratospheric $O_3$ from these two simulations, analyze this change in the context of dynamical-driven variabilities in $O_3$ before concluding such change is significant and the related aqueous process need to be considered in chemical modeling.

iii) By looking at Figures 8, 13, 14, with the documented ozone abundance and variability in previous literature, the change in tropospheric and lower stratospheric $O_3$ are too small, in my opinion, to suggest VOC emissions from Indonesian fires can change atmospheric $O_3$ in a significant way, except in the small localized regions near Indonesia.

Following the concerns of both referees, we significantly revised the section on the lower stratospheric composition. Originally, we obtained the change of -0.38 DU by simply taking the difference of both long term simulations. This is possible since all simulations are performed using the Chemistry-Transport Model mode (QCTM mode, Deckert et al., 2011), meaning that chemistry and dynamics are decoupled. This is achieved by using fixed tracer mixing rations as input for the radiation scheme instead of the prognostic chemical tracers. In this way, the meteorology is the same (binary identical) for all simulations and therefore all changes in the atmospheric chemical composition predicted by EMAC are solely due to the additional VOC emissions from biomass burning. Thus, the differences in the lower stratospheric $O_3$ are not due to dynamical responses (which have been neglected in this study). The information that all simulations are used in QCTM mode was added to Sect. 2.2 in the revised manuscript.

Due to the comment of the other referee, we decided to remove this trend analysis and the related claims from the revised manuscript. Even though we agree that the changes in lower stratospheric $O_3$ are small and localised, we still think that it is an interesting finding that aromatic emissions from the Indonesian fires potentially contribute to the lower stratospheric $O_3$ variability. Therefore, we now focus on the change in the importance of the $O_3$ loss by phenoxy radicals in the UTLS and its potential to contribute to the lower stratospheric $O_3$ variability. We find that in the upper northern tropical troposphere, this loss pathway contributes up to 40 % to the total chemical $O_3$ loss. When aromatic emissions from the Indonesian peatland fires are considered, this contribution further increases in the upper southern tropical troposphere by up to 20 % (from about 20 % to 40 %). Additionally, we broadened the discussion on the lower stratosphere, which now follows the same structure as the tropospheric analysis (hydrocarbons, oxygenated organics, nitrogen containing compounds, and radicals).

Concerning your comment on the lower tropospheric changes in $O_3$: We completely agree that the changes in $O_3$ on a global scale are insignificant (except for Boreal Asia in 2003), as stated in the manuscript. Following your comment, we revised the tropospheric $O_3$ part and removed Fig. 8 in the revised version.

**Minor comments**

L12. hydroxyl radicals (OH) $\rightarrow$ the hydroxyl radical

Done.

L13-14. Suggested rewording: While an increase in ozone is predicted close to the peatland fires, simulated O$_3$ decreases in eastern Indonesia due to particularly high phenols.

We followed your suggestion but changed the last part to "[...] due to particularly high phenol concentrations.".

L16. "the impact of such extreme pollution events." The impact on what? Please clarify.

In the revised manuscript this now reads: " [...] of such extreme pollution events on the atmospheric composition.".

L18. Not really appropriate to use "high" destruction. May be substitute it with "large" or "efficient"?

Please note that this statement is not longer included in the revised manuscript.

L20-22. What's the statistical significance of such decrease compared to the variability of ozone in this region? See above major comment.

Following your comment and the comment of the other referee, we removed the trend analysis on lower stratospheric O$_3$. Therefore, please note that this statement is no longer included in the revised manuscript.

L26. "," missing before "resulting"

Done.

L30. Please specify which months are the dry season.

In 2015, the dry season started in mid June and lasted until November (Field et al., 2016). This is now included in the revised manuscript.

L33. → Gaveau et al. (2014) estimated

Done.

L39. Delete "this" before "smouldering fires"

Done.

L41. → A major fraction...is (instead of "are")

Done.

L41. → which comprise a large variety of species and can influence atmospheric...

Done.

L48. "This almost stationary globally prevailing meteorological pattern" – too many adjectives! And it is not global either. How about "This semi-stationary large-scale meteorological pattern"?

We adjusted the sentence following your suggestions.

L58. What do you mean by "usual systems"? Consider use "other meteorological systems"

We included your recommendation in the revised manuscript.

L76. → where the most of peatland is burned.

We changed this. Please note that following your earlier comment, this part was moved to another section (now Sect. 3).

L76. How about change "result in" to "have"?

This now reads: "Since non-peatland biomass burning fuels have lower VOC emissions [...]".

L77. "Indonesia is characterized by a unique emission footprint". It is helpful to readers if you can elaborate the uniqueness of the Indonesian fire emissions, instead of implying it is unique. What's unique about it? You may consider move the sentence from L83-84 to here.

In the revised manuscript, an elaboration was included about the unique emission footprint of the Indonesian peatland fires (high VOC and high aromatic emissions). Due to the restructuring of the manuscript, this discussion is now included in Sect. 3.

L103. Suggest use "simulated results with", instead of "prediction to..."

Done.

L109. → provided

Done.

L119. Within this... → For this...

Done.

L119. Suggest change "modelled" to "computed" to avoid redundancy with submodels

Done.

L125. You may consider specify "linear hydrocarbons" for readers that are not familiar with this term

This now reads :" [...] and anthropogenic aliphatic and aromatic hydrocarbons."

L127. I would suggest use "primarily emitted" instead of "heavily emitted"

Done.

L150. It is not 100 % accurate to say biogenic and biomass burning emissions are the dominant sources of VOCs, consider the importance of anthropogenic emissions. How about change to "two major sources..."

This is indeed correct. We changed it accordingly in the revised manuscript.

L152. "MESSy submodel Model of..." reads awkward and redundant. How about use "The MESSy submodel uses Model of..."

Done.

L243. Delete "even"

Done.

L399. Change "$O_3$ sinks" to "an $O_3$ sink"

Done.

**References**

Deckert, R., Jöckel, P., Grewe, V., Gottschaldt, K.-D., and Hoor, P.: A quasi chemistry-transport model mode for EMAC, Geoscientific Model Development, 4, 195–206, https://doi.org/10.5194/gmd-4-195-2011, 2011.

Field, R. D., van der Werf, G. R., Fanin, T., Fetzer, E. J., Fuller, R., Jethva, H., Levy, R., Livesey, N. J., Luo, M., Torres, O., and Worden, H. M.: Indonesian fire activity and smoke pollution in 2015 show persistent nonlinear sensitivity to El Niño-induced drought, Proceedings of the National Academy of Sciences, 113, 9204–9209, https://doi.org/10.1073/pnas.1524888113, 2016.

Gaveau, D. L. A., Salim, M. A., Hergoualc'h, K., Locatelli, B., Sloan, S., Wooster, M., Marlier, M. E., Molidena, E., Yaen, H., DeFries, R., Verchot, L., Murdiyarso, D., Nasi, R., Holmgren, P., and Sheil, D.: Major atmospheric emissions from peat fires in Southeast Asia during non-drought years: evidence from the 2013 Sumatran fires, Scientific Reports, 4, 6112, https://doi.org/10.1038/srep06112, 2014.

---

## Author Comment (AC2) · 12 Apr 2021

**Reply to comments of Anonymous Referee #2**

This paper presents modeling studies of several aspects of the strong Indonesian biomass burning in 2015, including the impact of VOC emissions, the in-cloud oxidation of VOCs, and the transport of VOCs to the lower stratosphere by the Asian monsoon.This work uses the ECHAM/MESSy model for a series of sensitivity tests, including two horizontal resolutions (1 deg and 2.8 degrees), and a comparison of aqueous-phase mechanisms. A new satellite product, HCN from IASI, is described in the appendix and used for model evaluation. A model experiment where all VOC biomass burning emissions are excluded is the basis for much of the analysis in the paper. There area number of interesting results and new results, such as the aqueous chemistry impacts, presented, but I agree with the other reviewer that a major revision of the paper is required before publication.

Thank you very much for the helpful comments and seeing the value of our work to the community. Following the comments of both referees, we revised the manuscript significantly focusing on the following points:

- The title of the manuscript was adjusted to better represent the structural changes performed during the revision.

- We moved the comparison of the emission footprint of the Indonesian peatland fires to other biomass burning regions from the introduction to a new dedicated section (now Sect. 3).

- In addition to updating the section on the comparison of EMAC's representation of HCN to IASI satellite retrievals, we now also include a comparison to IASI CO retrievals.

- We restructured the reporting of our findings. It now follows a more systematic manner, in which we first report the impact on the troposphere (now Sect. 5), followed by the impact on the lower stratosphere (now Sect. 6). In both sections, we now focus on hydrocarbons, oxygenated organics, nitrogen containing compounds, and radicals in separate subsections. Afterwards the impact of in-cloud OVOC oxidation on the changes in the troposphere and stratosphere is discussed in a separate section (now Sect. 7).

- We extended the tropospheric analysis to include further hydrocarbons and OVOCs.

- Following the concerns of both referees, we removed the trend analysis on the influence of the Indonesian peatland fires on lower stratospheric $O_3$. Instead, we broadened the discussion on the lower stratosphere (hydrocarbons, oxygenated organics, nitrogen containing compounds, and radicals). In addition, we now focus on the change in the importance of the $O_3$ loss by phenoxy radicals in the UTLS and its potential to contribute to the lower stratospheric $O_3$ variability.

Please find in black the original comments and in red our replies.

The paper includes a broad set of topics - tropospheric composition, health impacts,aqueous chemistry, UTLS impacts - which make the paper seem a bit disjointed.

We understand that the original structure of the manuscript might seem a bit disjoint. Based on the comments, we revised the structure of the result sections and changed it such that it follows a more systematic structure. We first report the impact on the troposphere (now Sect. 5), followed by the impact on the lower stratosphere (now Sect. 6). In both sections, we focus on hydrocarbons, oxygenated organics, nitrogen containing compounds, and radicals. Afterwards, the impact of in-cloud OVOC oxidation on the tropospheric and stratospheric composition is discussed in a separate section (now Sect. 7).

As the other reviewer pointed out, the impact on stratospheric ozone appears to be in-significant and I recommend leaving that out.

Following the concerns of both referees, we significantly revised the section on the lower stratospheric composition. In the revised manuscript, we decided to remove the performed trend analysis and the related claims for the total $O_3$ loss in the lower stratosphere. Even though we agree that the changes

in lower stratospheric $O_3$ are small and localised, we still think that it is an interesting finding that aromatic emissions from the Indonesian fires potentially contribute to the lower stratospheric $O_3$ variability. Therefore, we now focus on the change in the importance of the $O_3$ loss by phenoxy radicals in the UTLS and its potential to contribute to the lower stratospheric $O_3$ variability. We find that in the upper northern tropical troposphere, this loss pathway contributes up to 40 % to the total chemical $O_3$ loss. When aromatic emissions from the Indonesian peatland fires are considered, this contribution further increases in the upper southern tropical troposphere by up to 20 % (from about 20 % to 40 %). Additionally, we broadened the discussion on the lower stratosphere, which now follows the same structure as the tropospheric analysis (hydrocarbons, oxygenated organics, nitrogen containing compounds, and radicals).

The paper discusses VOCs quite generally in many places, but does briefly summarize in Section 4.1 various categories of VOCs, and describes their differences. A bit more focus is placed on aromatics, and particularly phenols, which are certainly significant in biomass burning, but a number of other HCs and OVOCs are also important and should be discussed.

Following your recommendation, we expanded the discussion of hydrocarbons and OVOCs.

The discussion of HCN does not really fit with the reactive chemistry analysis and I do not find the use of it to evaluate the biomass burning emissions in the model very convincing. This paper seems to be an opportunity to describe the new IASI retrievals of HCN, but it seems like that discussion might be appropriate for a short AMT paper, with comparisons to other satellite retrievals of HCN and validation with insitu aircraft observations. Evaluation of the model with the established and validated CO satellite retrievals, from IASI and other platforms, would provide greater confidence in the model simulations.

As explained in the manuscript, the HCN satellite data have been retrieved from IASI observations using a general framework that has been fully described in dedicated publications and that has already been applied successfully to the retrieval of ammonia and of several VOCs (we provide the references in the text). Therefore, describing again the entire methodology, e.g., in an AMT paper, would be redundant with previous works, and we believe it is more appropriate to limit the explanation in an appendix to the few elements specific to the HCN retrievals (e.g., spectral range, post-filtering). A comparison of the IASI HCN columns with independent measurements is foreseen as part of a general study dedicated to the validation of the different IASI VOC products. The 2015 Indonesian peatland fires led to substantial emissions of VOCs, in particular of nitrogen-containing species. As shown with the satellite measurements in Fig. A3, it is particularly true for HCN, during this event by far the highest HCN columns and the largest plume captured on record throughout the IASI operational time series (since late 2007). We therefore believe HCN is an interesting tracer to represent the exceptional amplitude and extent of these peatland fires.

On the other hand, we agree that CO is a typical, widely used biomass burning tracer. Therefore, following your suggestion and the comment of the other referee, we revised the HCN comparison and extended the analysis to include also a comparison of EMAC's CO to IASI retrievals in Sect. 4. Here, we find that EMAC tends to slightly underestimate CO over Indonesia and overestimates CO in South America. We attribute the underestimation in Indonesia to a too low emission coefficient used by EMAC. The overestimation in South America is related to an overestimation of biogenic emissions in this region. Following both analysis we are certain that EMAC represents the Indonesian peatland fires in a reasonable manner, especially when considering the exceptional strength of the 2015 Indonesian peatland fires.

**Here are some of the technical corrections that are needed:**

l.110: in the resolution label, L90 apparently indicates 90 levels; explain 'MA'.

Exactly, L90 indicates that 90 levels are used. Here, MA indicates the Middle Atmosphere (MA) version of the model, i.e. that the 90 levels focus on the lower and middle atmosphere. In the original submitted version this is explained in line 114 and 115. In the revised version, we adjusted the explanation of MA in L90MA.

l.123: 'capable to represent' should be 'capable of representing'

Done.

l.193: 'one simulation exists, in which all ...' → ' in one simulation all ...'

Done.

l. 339: 'destruct' → 'destroy'

Done.

l.348: "80's" → "1980s"

Done.

l.396: relative → relatively

Done.

l.405: rewrite "which even enhances by in-cloud ..."

Done.

l.407: 'in the same order to SOA formation" → "to SOA formation on the same order"

Done.

l.431: reference for La Nina strengthening AMSA?

Basha et al. (2020) report a higher extend of the AMSA during La Niña. Please note that this statement is not longer included in the revised manuscript.

l.431: 'strengthens' → 'has strengthened'

Please note that this statement is not longer included in the revised manuscript.

l.433: extend → extent

Please note that this statement is not longer included in the revised manuscript.

l.456: 'appoint' is not the right word here. suggest? hypothesize?

Please note that this statement is not longer included in the revised manuscript.

l.467: 'our study' - does this refer to this manuscript, or previous work? If this paper,where is this shown?

This refers to this manuscript. We do not show these findings in any figure. We now include "(not shown)" in the revised manuscript to clarify this for the reader.

l.506: deletion → depletion

Done.

Table 1: column 2 would be better labeled 'Dominant fuel type' (fire type implies to meflaming or smoldering, for example).

A very good point. In Table 1, column 2 is now referred to it as dominant fire type. Within the text, we also now refer to 'fuel type'.

Table 2: What does 'ScSta' mean? Also, define JAMOC here.

Here, ScSta refers to EMAC's standard aqueous-phase mechanism and JAMOC to the complex in-cloud OVOC oxidation scheme Jülich Aqueous-phase Mechanism of Organic Chemistry. We added an appropriate elaboration to the caption of Table 2 and refer to Sect. 2.1.1. for further details.

**References**

Basha, G., Ratnam, M. V., and Kishore, P.: Asian summer monsoon anticyclone: trends and variability, Atmospheric Chemistry and Physics, 20, 6789–6801, https://doi.org/10.5194/acp-20-6789-2020, 2020.

---

## Author Response (AR2)

Dear Bryan N. Duncan,

we are glad to hear that the revised version of the manuscript adequately addressed the concerns of the reviewers. We are thankful for the comments of Anonymous Referee #1 and think that our resulting modifications further improve the manuscript. Please find in black the original comments of the Anonymous Referee #1 and in red our reply:

1. Section 3. I suggest the authors consider adding a bit more quantitative emissions numbers in this section, instead of only have general terms such as "highest …, lowest …, about one third … less than half …". This change, albeit small, will make the paragraph much more informative.

   This is a very good point. We agree that adding quantitative emission numbers to this section will increase its value to the manuscript. In the revised version of this manuscript, quantitative emission numbers were thus added to section 3.

2. L236. Change to "et al., 2000; 2009"

   Done.

3. Sections 5 and 6: Shouldn't there also be two subsections 5.5 and 6.5 titled "O3"? In this revised version, the discussion on O3 is following the "Radicals" subsection, as if they are part of the radicals.

   We added separate ozone subsections to section 5 and 6.

At this point, we would like to thank you for serving as editor of this manuscript. In addition, we would like to thank all referees for their contributions.

Kind regards,
On behalf of the authors,

Simon Rosanka